# Characterization of novel phage Henuyfy11N: a potential therapeutic agent against extended-spectrum β-lactamase (ESBL)-producing *Escherichia coli*

Xinwei Zhang,[1,2] Jiaqi He,[1,2] Dongliang Qiao,[1,2] Qiming Li,[1,2] Zhigang Liu,[1,2] Li Wang[1,2,3]

**ABSTRACT** *Escherichia coli* is a major cause of hospital-acquired infections in China, including urinary tract, bloodstream, and intestinal infections. Given the rising prevalence of antibiotic-resistant *E. coli*, phages are increasingly regarded as promising alternatives to conventional antibiotics. Henuyfy11N was isolated using the double-layer agar method and characterized via transmission electron microscopy in this study. Biological assays included stability under varying pH and temperature, UV sensitivity, host range, optimal multiplicity of infection, adsorption rate, and one-step growth curve. *In vitro* lytic activity against extended-spectrum β-lactamase (ESBL)-producing *E. coli* and biofilm eradication capacity was assessed. Whole-genome sequencing enabled phylogenetic, synteny (the analysis of conserved blocks of genetic sequence between different genomes), and functional annotation analyses. *In vivo*, the therapeutic efficacy was evaluated in a mouse infection model. Phage Henuyfy11N has not yet been classified. It demonstrated high lytic activity, a short latent period, and a burst size of 57.1 PFU/cell. The phage remained stable across a broad pH range (3–11) and temperatures up to 70°C. Its circular double-stranded DNA genome (41,103 bp, G + C% 50.88) contains 54 open reading frames, with no tRNA, virulence, or antibiotic resistance genes. Genomic and phylogenetic analyses revealed close relatedness to phage BUCT789. Henuyfy11N effectively lysed ESBL-producing *E. coli*, disrupted biofilms, and significantly improved survival in the mouse infection model. Henuyfy11N shows high host specificity, efficient lytic activity, rapid replication, and a safe genomic profile, demonstrating some potential as a therapeutic agent against ESBL-producing *E. coli* infections.

**IMPORTANCE** The widespread use of antibiotics has led to increasing antibiotic resistance, which is a growing global health concern. Therefore, the development of novel antimicrobial therapy that can cure drug-resistant bacteria-induced infections is imperative. Phages are of increasing interest as natural enemies of bacteria, with clear advantages in antibacterial applications. In this study, by using extended-spectrum β-lactamase (ESBL)-producing *Escherichia coli* 2025011N as a host, we successfully isolated and purified *Escherichia* phage Henuyfy11N and conducted a series of experiments to verify its genomic character and biological character. Our findings revealed that the phage exhibited excellent tolerance to a broad spectrum of pH and wide temperature range. Phage Henuyfy11N was effective in disrupting mature biofilm, and no genes for virulence, lysogenic, integrase, or AMRs were found in the genome. Besides, Henuyfy11N showed promising antibacterial effects *in vivo* and *in vitro*, indicating potential as a therapeutic agent against ESBL-producing *E. coli* infections.

**KEYWORDS** extended-spectrum β-lactamase, *Escherichia* phage, phage therapy

**Peer Reviewer** Saisubramanian Nagarajan, Shanmugha Arts Science Technology and Research Academy School of Chemical and Biotechnology, Thanjavur, Tamil Nadu, India

Address correspondence to Li Wang, wangli851217@163.com.

The authors declare no conflict of interest.

See the funding table on p. 18.

*Escherichia coli* is a common pathogen responsible for both community- and hospital-acquired infections (1–3). The increasing prevalence of antimicrobial resistance (AMR) in *E. coli* has led to rising mortality rates associated with resistant infections (4–7). The overuse and misuse of broad-spectrum antibiotics, especially third-generation cephalosporins, have driven the emergence of extended-spectrum β-lactamase (ESBL)-producing *E. coli* (8). Since the beginning of this century, ESBLs have been recognized as a serious public health threat (9, 10). In Nepal, the prevalence of ESBL-producing *E. coli* increased from 18.7% in 2011 to 29.5% in 2021 (11). Similarly, in Chinese hospitals, the isolation rate of ESBL-producing *E. coli* reached 52.6% in 2021 (12). Common antimicrobial agents used against ESBL-producing *E. coli* include tetracyclines, β-lactams, fluoroquinolones, aminoglycosides, and sulfamethoxazole-trimethoprim combinations. However, the marked rise in AMR among *Enterobacteriaceae* in recent years has significantly limited therapeutic options. Although ESBLs can hydrolyze most β-lactam antibiotics, they remain ineffective against carbapenems. Consequently, carbapenems are currently among the few effective treatments available for infections caused by ESBL-producing *Enterobacteriaceae* (13, 14). Nevertheless, the overuse of carbapenem antibiotics for treating ESBL-producing *E. coli* has contributed to a growing number of infections caused by carbapenemase-producing organisms. Once a patient is infected with carbapenem-resistant bacteria, very few antimicrobial agents remain available for clinical use (15, 16).

As the detection rate of carbapenem-resistant bacteria continues to rise, tigecycline, polymyxin, and ceftazidime/avibactam have become crucial therapeutic options for infections caused by these pathogens (17). However, monotherapy is often limited in efficacy. While dose escalation may be considered when standard regimens fail, it can lead to increased toxicity and a higher risk of selecting for resistance. Moreover, given the considerable time and cost associated with developing novel antibiotics, there is an urgent need to implement rational strategies that enhance the effectiveness of existing agents and delay the development of resistance to these last-line drugs. It is essential to preserve the activity of tigecycline, polymyxin, and ceftazidime/avibactam to maintain effectiveness against carbapenem-resistant bacteria. Consequently, innovative approaches and alternative treatment modalities are urgently required to combat these emerging multidrug-resistant pathogens.

Phage therapy presents several potential advantages over conventional antibiotic treatments. Unlike broad-spectrum antibiotics, phages exhibit high host specificity, selectively targeting pathogenic bacteria without significantly disrupting the commensal microbiota, thereby reducing the risks of ecological imbalance and secondary infections. Moreover, the development of resistance to phages is generally considered to be less likely than to antibiotics, owing to their co-evolutionary dynamics and narrow target range. Phage therapy is also characterized by a favorable safety profile, with no severe side effects reported in clinical settings to date. Furthermore, the development cycle for phage-based therapeutics is substantially shorter and more cost-effective than that for novel antibiotics. While antibiotic discovery can take years, the isolation and characterization of new phages can typically be accomplished within days or weeks. These attributes have established phage therapy as a promising strategy against multidrug-resistant bacterial infections and an emerging focus of contemporary research (18, 19).

This study aimed to isolate and characterize bacteriophages capable of lysing ESBL-producing *E. coli*. The stability under various environmental conditions and molecular biological features of the isolated phages were investigated to evaluate their potential as alternatives to conventional antibiotics. Furthermore, the lytic efficacy against planktonic host bacteria and antibiofilm activity were assessed *in vitro*. Finally, the therapeutic potential of the phage was preliminarily validated using a mouse infection model, providing experimental support for the use of phage therapy against infections caused by ESBL-producing *E. coli*.

## MATERIALS AND METHODS

### Bacterial strains and cultures

*E. coli*, *Klebsiella pneumoniae*, *Acinetobacter baumannii*, and *Pseudomonas aeruginosa* isolates were grown in lysogeny broth (LB) broth (Solarbio, Beijing, China) in a shaking incubator set at 220 rpm and 37°C. All the isolates were identified through 16S rRNA sequencing using universal primers 27F (5′-AGAGTTTGATCCTGGCTCAG-3′) and 1492R (5′-TACGGCTACCTTGTTACGACTT-3′). Phage Henuyfy11N was isolated from the sewage well of the First Affiliated Hospital of Henan University. Phage Henuyfy11N can be obtained by contacting the China Center for Type Culture Collection or the corresponding author.

### Isolation and purification of phage Henuyfy11N

The phage was isolated from untreated wastewater obtained from the sewage well at the First Affiliated Hospital of Henan University. The sample was centrifuged at $12,000 \times g$ for 10 min, and the supernatant was filtered through a 0.22 µm membrane (Lige Science, Guangzhou, China). The host bacterium, previously stored at −80°C, was cultured in LB broth (Solarbio) to logarithmic phase ($OD_{600} \approx 0.5$). The filtered wastewater was added to the bacterial culture and co-incubated for 12–16 h at 37°C. After centrifugation and filtration, the presence of the phage was detected via a double-layer agar plaque assay, showing clear plaques. For purification, a single plaque was picked and inoculated into log-phase bacterial culture for amplification. The process of plaque picking, amplification, and double-layer agar assay was repeated at least five times to obtain a purified phage population. The final lysate was centrifuged, filtered through a 0.22 µm membrane, and titrated to ensure a concentration above $10^{10}$ PFU/mL before storage at 4°C.

### Transmission electron microscopy of phage particles

Purified phage particles with a titer of at least $10^{10}$ PFU/mL were used for morphological analysis by transmission electron microscopy (TEM). Approximately 20 µL of the phage suspension was applied onto a carbon-coated copper grid and allowed to adsorb for 30 min at room temperature in the dark. The grid was then negatively stained with 1% (wt/vol) phosphotungstic acid (pH-adjusted to 7.0) for 20 min. Excess stain was carefully removed using filter paper, and the grid was air-dried for 30 min. Phage morphology was examined under a transmission electron microscope (JEOL JEM-2100F, Tokyo, Japan) operated at an accelerating voltage of 80 kV. Images were captured at suitable magnifications to visualize and measure the phage particles.

### Host range of phage Henuyfy11N

The host range of the purified phage was determined against a panel of clinical isolates, including 150 strains of ESBL-producing *E. coli*, 13 strains of *Acinetobacter baumannii*, and 3 strains of *Pseudomonas aeruginosa*. Each bacterial strain was cultured individually until the $OD_{600}$ reached approximately 0.5, indicating the logarithmic growth phase. Then, 0.5 mL of each bacterial suspension was mixed with 5 mL of the molten LB semisolid agar and overlaid onto LB solid agar plates. The plates were allowed to solidify at room temperature. The concentrated phage stock was serially diluted 10-fold in sterile phosphate-buffered saline (PBS) containing $MgCl_2$ and $CaCl_2$. Subsequently, 5 µL of each dilution was spotted onto the surface of the prepared double-layer agar plates corresponding to each bacterial strain. After the spots dried and were absorbed, the plates were incubated overnight at 37°C. The following day, the plates were examined for the presence of clear zones or individual plaques at the spotting sites. A strain was considered susceptible if lytic spots or plaques were observed, indicating successful phage infection and lysis. The efficiency of plating (EOP) was calculated for susceptible hosts by comparing the plaque counts to those obtained on the primary host.

## Efficiency of plating

Bacteriophage Henuyfy11N was tested three times against all its host bacteria using three different dilution levels. All bacterial strains were cultured overnight (18 h) at 37°C, with 200 µL of each culture combined with 100 µL of diluted phage lysate for double-layer plaque assays. The three preparations of bacteriophage Henuyfy11N were diluted from the original phage stock at ratios of 101–103. This approach allowed for the parallel execution of EOP experiments with specific phages on all tested bacterial strains at concentrations comparable to those used in field tests. After overnight incubation at 37°C, the number of plaque-forming units (PFU) for each combination was counted. Finally, the EOP (average PFU of target bacteria/average PFU of host bacteria) and the standard deviation of the three measurements were calculated: EOP = (average PFU on target bacteria) / (average PFU on host bacteria) (20, 21).

## Environmental stability assays of phage Henuyfy11N

The stability of the purified phage was evaluated under various environmental conditions, including temperature, pH, and UV exposure. For thermal stability, aliquots of phage suspension were incubated at different temperatures (4°C, 10°C, 20°C, 30°C, 40°C, 50°C, 60°C, 70°C, and 80°C) for 1 h, followed by immediate cooling on ice, and the surviving phage titer was determined using the double-layer agar assay. For pH stability, LB media were adjusted to pH values ranging from 1 to 12 using HCl or NaOH, mixed with the phage solution, and incubated at 37°C for 1 h before neutralization and titration. UV sensitivity was assessed by exposing the phage suspension to UV light (110 µW/cm² at a 50 cm distance). Samples were collected at 0, 10, 20, 30, 40, 50, and 60 min, serially diluted, and plated for viable count. All assays were performed in triplicate.

## Phage adsorption assay

The adsorption rate of phage Henuyfy11N to its host, *E. coli* 2025011N (ECO11N), was determined as follows. A fresh bacterial culture was prepared to a concentration of approximately $10^{10}$ CFU/mL. Then, 1 mL of this culture was mixed with 10 µL of phage suspension ($10^{10}$ PFU/mL) to achieve a multiplicity of infection (MOI) of 0.01. The mixture was supplemented with pre-warmed fresh LB broth and incubated with shaking at 160 rpm and 37°C. Samples (100 µL) were collected at time intervals of 0, 2, 4, 6, 8, and 10 min post-infection. Each sample was immediately centrifuged at $10,000 \times g$ for 1 min to sediment bacterial cells and adsorbed phages. The supernatant was filtered through a 0.22 µm membrane filter to remove any remaining cells. The filtrate, containing unabsorbed free phages, was titrated using the double-layer agar method. The phage titer at each time point was used to calculate the percentage of non-adsorbed phages relative to the titer at time zero. The adsorption rate was derived based on the decrease in free phage concentration over time. All experiments were performed in biological triplicate. The adsorption rate constant for phage Henuyfy11N was calculated using the following equation (22):

$$k = -\ln(P/P_0)/\mathrm{Bt},$$

where $k$ is the adsorption rate constant (mL/min); $P$ is the free phage concentration per milliliter; $P_0$ is the initial phage concentration; $B$ is the initial bacterial density; and $t$ is the time (min).

## Determination of optimal multiplicity of infection

To determine the optimal MOI for bacteriophage Henuyfy11N, the following experimental procedure was employed. First, an overnight culture of the host bacterium *E. coli* ECO11N was diluted in fresh lysogeny broth (LB) medium and incubated with shaking at 37°C until the mid-logarithmic growth phase was reached ($OD_{600} \approx 0.5$). The bacterial cells were then harvested by centrifugation and resuspended in fresh LB to standardize

the concentration. Aliquots of this bacterial suspension were mixed with serial dilutions of a purified bacteriophage stock to achieve target MOI values of 0.001, 0.01, 0.1, 1, 10, and 100, based on the estimated number of bacterial cells. The phage-bacteria mixtures were incubated with vigorous shaking (220 rpm) at 37°C for 16 h to allow for complete phage replication and host cell lysis. After incubation, the cultures were centrifuged to pellet cellular debris, and the supernatants were filtered through a 0.22 µm membrane filter to remove any remaining bacteria. The filtrates were then serially diluted in sterile PBS containing $MgCl_2$ and $CaCl_2$, and the phage titers (PFU/mL) were quantified using the double-layer agar plaque assay. The MOI that resulted in the highest phage titer was identified as the optimal multiplicity of infection for subsequent experiments.

## One-step growth curve assay

The one-step growth curve of phage Henuyfy11N was characterized using its host strain *E. coli* ECO11N as previously reported with some modifications (23). Briefly, a bacterial culture was grown to mid-log phase ($OD_{600} \approx 0.5$), and 1 mL of this culture was mixed with 1 mL of phage suspension at an MOI of 0.01. The mixture was incubated at 37°C for 10 min to allow phage adsorption. To remove unabsorbed phages, the sample was centrifuged at $5,000 \times g$ for 5 min; the supernatant was discarded; and the pellet was resuspended in 1 mL of sterile LB medium. This resuspension was then transferred into 20 mL of fresh pre-warmed LB broth and incubated at 37°C with shaking. Samples (200 µL) were collected at 0, 10, 20, 30, 40, 50, 60, 70, 80, 90, 100, 110, and 120 min post-infection. Each sample was immediately centrifuged at $12,000 \times g$ for 2 min to separate cells from the supernatant. The supernatant (100 µL) was serially diluted in sterile PBS containing $MgCl_2$ and $CaCl_2$, and phage titers at each time point were determined using the double-layer agar method. The entire experiment was performed in triplicate. The one-step growth curve was plotted with time on the *x*-axis and phage titer (log10 PFU/mL) on the *y*-axis. The burst size was calculated as the ratio of the phage titer at the end of the rise period to the initial number of infected host cells, yielding an average burst size of 57.1 PFU/infected cell: burst size = (total progeny at final plateau) / (number of infected cells at initial plateau) (24).

## Analysis of bacteriolytic activity at different MOIs

The inhibitory effect of phage Henuyfy11N on the growth of its host bacterium, *E. coli* ECO11N, was assessed by monitoring culture optical density over time at different MOIs as previously described (25). A single colony of ECO11N was inoculated into 10 mL of fresh LB broth and incubated at 37°C with shaking until the culture reached the mid-exponential growth phase ($OD_{600} \approx 0.5$). This bacterial suspension was aliquoted into separate flasks and infected with the phage suspension to achieve target MOIs of 0.01, 0.1, 1, and 10. A control group, containing an equal volume of LB broth instead of phage suspension, was included. All experimental and control groups were incubated at 37°C with shaking. The $OD_{600}$ of each culture was measured at 30-min intervals over a total period of 10 h. The experiment was performed in triplicate. The lytic activity of the phage was evaluated by comparing the growth curves ($OD_{600}$ vs time) of the infected cultures to the phage-free control.

## Biofilm inhibition and removal assay

The ability of phage Henuyfy11N to inhibit biofilm formation and eradicate pre-existing biofilms was evaluated *in vitro* using a 96-well plate model (26). For the biofilm inhibition assay, a diluted suspension of *E. coli* ECO11N (100-fold dilution of a mid-log phase culture, $OD_{600} \approx 0.5$) was added to wells (200 µL per well) and immediately supplemented with phages at MOIs of 0.01, 0.1, 1, and 10. A phage-free well served as the negative control. After 24 h of incubation at 37°C, non-adherent cells were removed by washing five times with PBS. The plates were dried at 65°C for 1 h, and biofilms were stained with 1% crystal violet (Solarbio) for 30 min. Excess stain was removed by

rinsing with distilled water, and the plates were air-dried. Bound dye was solubilized with 95% ethanol, and absorbance was measured at 570 nm. For the biofilm removal assay, biofilms were pre-formed by incubating the bacterial suspension for 24 h at 37°C. Thereafter, the phage was added at the same range of MOIs and incubated for another 24 h. The remaining biofilm biomass was quantified using the same crystal violet staining protocol. All experiments were performed in triplicate.

## Extraction of phage genomic DNA

Genomic DNA of phage Henuyfy11N was extracted from high-titer lysates using a modified enzymatic and chemical lysis protocol (25). Briefly, the host strain *E. coli* ECO11N was grown in 50 mL of LB broth at 37°C to an $OD_{600}$ of approximately 0.5. Then, 100 µL of purified phage stock ($\geq 10^{10}$ PFU/mL) was added to the culture and incubated for 8 h at 37°C with shaking. A chloroform substitute (Beyotime, Shanghai, China) was added to the lysate at a final concentration of 0.1%–0.5% and incubated at 37°C for 5-10 min to complete bacterial lysis. The lysate was centrifuged at 10,000 × *g* for 20 min at 4°C to remove cellular debris. The supernatant was treated with 5 µL of RNase A and 10 µL of DNase I (Yuanye, Shanghai, China) and incubated at 37°C for 30 min to degrade nucleic acids from host cells. Subsequently, 4 mL of phage precipitation solution (containing 20% polyethylene glycol and 2.5 M sodium chloride) (Yuanye) was added and mixed thoroughly, and the sample was incubated on ice for 1 h or overnight at 4°C. Precipitated phage particles were collected by centrifugation at 10,000 × *g* for 20 min at 4°C. The pellet was resuspended in 1 mL of SM buffer (Yuanye), transferred to a clean tube, and mixed with 40 µL of phage lysis buffer (Yuanye), followed by incubation at 68°C for 15 min. An equal volume of protein removal buffer (Yuanye) was added, and the mixture was gently inverted before centrifugation at 12,000 × *g* for 5 min. The aqueous phase was transferred to a new tube, mixed with an equal volume of pre-chilled phage wash solution (Yuanye), and incubated at −20°C for 1 h. After centrifugation at 12,000 × *g* for 10 min at 4°C, the supernatant was discarded. The DNA pellet was washed with 70% ethanol, centrifuged at 8,000 × *g* for 8 min at 4°C, and air-dried at room temperature. The purified DNA was resuspended in an appropriate volume of TE buffer and stored at −20°C.

## Genome sequencing, annotation, and bioinformatics analysis

The extracted genomic DNA of phage Henuyfy11N described above was sent to Personalbio (Nanjing, China) for whole-genome sequencing utilizing Illumina technology (NovSeq X Plus, USA). Briefly, quality assessment of the raw sequencing data were performed with Fastp v0.20.0. Low-quality reads and adapter sequences were trimmed using AdapterRemoval v2.2.2 and SOAPec v2.03 under default parameters. The resulting high-quality reads were *de novo* assembled using SPAdes v3.12.0 to generate the complete genome sequence (27). Open reading frames (ORFs) were predicted using the Softberry online platform (https://www.softberry.com). Functional annotation of the predicted proteins was conducted via BLASTp (https://blast.ncbi.nlm.nih.gov/Blast.cgi) analysis against the NCBI non-redundant protein database (https://www.ncbi.nlm.nih.gov/). The fully annotated genome was deposited into the NCBI database, and the sequence was released under accession number PV696613.1. A genomic map was generated using CGView (28). Additionally, the PhageScope online platform was employed to identify tRNA and tmRNA genes, as well as to screen for potential antibiotic resistance genes and virulence factors (29). Genomic similarity among phages was quantified and visualized using heatmaps generated by VIRIDIC (30). Phage genomic multiple linear alignment diagrams were drawn using Easyfig software (31).

## Phylogenetic analysis

The evolutionary relationships between phage Henuyfy11N and other reference phages were investigated through whole-genome sequence and protein-based phylogenetic

analyses. Whole-genome comparisons were conducted using MegaBLAST to assess global sequence similarity. For more conserved protein-level phylogeny, amino acid sequences of key phage proteins—including DNA polymerase, terminase large and small subunits, and major capsid protein—were retrieved and aligned using ClustalW. Phylogenetic trees were reconstructed with the neighbor-joining method in MEGA 11 software, with branch support evaluated by 1,000 bootstrap replicates (32). Both genome-wide and protein-specific alignments were utilized to infer phylogenetic topology and evolutionary divergence.

## Determination of the minimum lethal dose of ECO11N in a mouse model

To establish the minimum lethal dose (MLD) resulting in 100% mortality, a total of 35 mice were randomly allocated into seven groups ($n$ = 5 per group). Each group received an intraperitoneal injection of a bacterial suspension of *E. coli* ECO11N at one of the following concentrations: $5.2 \times 10^{10}$, $2.6 \times 10^{10}$, $1.3 \times 10^{10}$, $6.5 \times 10^9$, $2.6 \times 10^9$, and $1.3 \times 10^9$ CFU per mouse). The control group was administered an equivalent volume of sterile PBS containing $MgCl_2$ and $CaCl_2$ via the same route. The survival status of the mice was monitored and recorded every 24 h post-injection.

## Evaluation of phage therapy with Henuyfy11N in a mouse infection model

To assess the therapeutic efficacy of phage Henuyfy11N against lethal *E. coli* ECO11N infection, all mice were randomly assigned to six groups ($n$ = 5 per group). Each mouse was first challenged intraperitoneally with the pre-determined MLD of *E. coli* ECO11N. Immediately following bacterial infection, mice in four treatment groups received an intraperitoneal injection of phage Henuyfy11N at MOIs of 0.01, 0.1, 1, or 10. A positive control group received sterile PBS containing $MgCl_2$ and $CaCl_2$ instead of phage treatment after bacterial challenge, while a negative control group received PBS containing $MgCl_2$ and $CaCl_2$ in place of both bacteria and phage. Survival was monitored every 24 h, and mortality was recorded for each group over the course of the experiment.

## Data statistical analysis

All data are presented as mean ± standard deviation. Statistical analysis was performed using SPSS v5 (IBM Corp, Armonk, NY, USA) and GraphPad Prism 9.0 (GraphPad Software, Inc., San Diego, CA, USA). The significance of differences between two groups was determined using unpaired $t$-tests, while the significance of differences among multiple groups was assessed using one-way ANOVA. Statistical significance was set at $P < 0.05$.

## RESULTS

### Morphological characterization of phage Henuyfy11N

Henuyfy11N was isolated from wastewater samples obtained from the First Affiliated Hospital of Henan University in Kaifeng City, China, using the double-layer agar method with single-plaque isolation. On lawns of its host strain *E. coli* ECO11N, the phage formed clear plaques with a needle-shaped morphology. (Fig. 1A). Notably, the plaque size remained consistent between 24 h of incubation. TEM revealed that phage Henuyfy11N exhibits a typical tailed phage morphology, featuring an icosahedral head measuring 74.23 ± 5.87 nm in diameter, a narrow collar region, and a long non-contractile tail with a length of 132.02 ± 2.49 nm at 50 nm resolution, while at 100 nm resolution, the phage capsid diameter and tail length were 75.34 ± 9.57 nm and 132.34 ± 7.13 nm (Fig. 1B and C). Based on the taxMyPhage analysis, which indicates that phage Henuyfy11N has not yet been classified by the International Committee on Taxonomy of Viruses (33, 33), it is highly likely that bacteriophage Henuyfy11N represents a novel genus and species (Fig. S1).

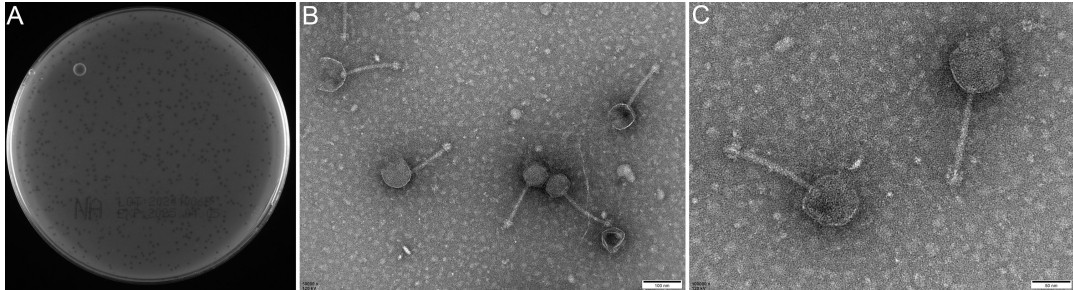

**FIG 1** Morphological characteristics of phage Henuyfy11N. (A) Plaque morphology of phage Henuyfy11N on a lawn of *E. coli* ECO11N after 24 h of incubation using the double-layer agar method. (B) TEM of Henuyfy11N negatively stained with 1% phosphotungstic acid, showing the icosahedral head and tail structure. The phage capsid diameter and tail length were 75.34 ± 9.57 nm and 132.34 ± 7.13 nm. Scale bar: 100 nm. (C) High-magnification TEM image illustrating the detailed structure of the phage head (74.23 ± 5.87 nm in diameter) and the contractile tail (132.02 ± 2.49 nm in length). Scale bar: 50 nm.

## Host range of phage Henuyfy11N

Henuyfy11N exhibited a narrow lytic spectrum among the clinical isolates tested. Out of 150 *E. coli* strains isolated from the First Affiliated Hospital of Henan University, only two were susceptible to lysis by Henuyfy11N: *E. coli* 2025011N (ECO11N) and *E. coli* 2021190N (ECO190N). No lytic activity was observed against any of the 13 strains of *A. baumannii* or the 3 strains of *P. aeruginosa* included in the panel (Table S1). ECO11N was isolated from a blood sample of a 63-year-old female patient in the department of neurology, while ECO190N originated from a urine sample of a 62-year-old female patient diagnosed with ureteral stones. Henuyfy11N exhibits an infection efficiency value (EOP) of 100% against its original host, ECO11N. In contrast, it demonstrates an even higher EOP of 144.7% against the host ECO190N. The antimicrobial susceptibility profiles of both susceptible strains are summarized in Table 1.

## Biological characteristics of Henuyfy11N

To characterize the infection dynamics of Henuyfy11N, we analyzed its one-step growth curve, optimal MOI, and adsorption rate. The optimal MOI was determined to be 0.01, resulting in a maximum phage titer of $3.6 \times 10^{11}$ PFU/mL (Fig. 2A). Adsorption assays indicated that more than 99% of phages were adsorbed to host cells within 10 min (Fig. 2B). The one-step growth curve revealed a latent period of less than 10 min, followed by a rise period of approximately 80 min. The average burst size was calculated to be approximately 57.1 PFU/infected cell (Fig. 2C). The latent period is defined as the interval between phage adsorption and the onset of host cell lysis and progeny release, while the burst size represents the ratio of phage titer at the end of the burst phase to the initial concentration of infected host cells.

## Temperature, UV, and pH stability of Henuyfy11N

The stability of Henuyfy11N under various physical and chemical conditions was evaluated to assess its potential for practical application. The phage remained highly stable at temperatures between 10°C and 40°C, with no significant loss of activity. A slight reduction in viability was observed at 50°C, while exposure to temperatures above 70°C led to a sharp decline in activity, eventually resulting in complete inactivation (Fig. 2D). Under ultraviolet radiation (110 µW/cm² at a 50 cm distance), phage activity decreased markedly within the first 10 min. After 40 min of exposure, the activity stabilized with no further reduction (Fig. 2E). Henuyfy11N also demonstrated broad pH stability, retaining high activity within the pH range of 4–11. However, under strongly acidic (pH ≤ 3) or strongly alkaline (pH ≥ 12) conditions, a significant loss of activity was observed, often leading to complete inactivation (Fig. 2F).

**TABLE 1** Antibiotic sensitivity of host bacteria and its determination criteria[a]

| | *E. coli* 2025011N | | | | *E. coli* 2021190N | | | |
|---|---|---|---|---|---|---|---|---|
| | Breakpoints (µg/mL) | Experimental method | Result (µg/mL) | Explanation of classification | Breakpoints (µg/mL) | Experimental method | Result (µg/mL) | Explanation of classification |
| Ampicillin | S ≤ 8, R ≥ 32 | MIC | ≥32 | R | S ≤ 8, R ≥ 32 | MIC | ≥32 | R |
| Cefazolin | S ≤ 16, R ≥ 32 | MIC | ≥8 | R | S ≤ 16, R ≥ 32 | MIC | ≥8 | R |
| Ampicillin/sulbactam | S ≤ 8, R ≥ 32 | MIC | 8/4 | S | S ≤ 8, R ≥ 32 | MIC | 2/1 | S |
| Ticarcillin/clavulanate | S ≤ 16, R ≥ 128 | MIC | 16/2 | S | S ≤ 16, R ≥ 128 | MIC | ≤4/2 | S |
| Piperacillin/tazobactam | S ≤ 16, R ≥ 128 | MIC | ≤4/4 | S | S ≤ 16, R ≥ 128 | MIC | ≤4/4 | S |
| Cefuroxime | S ≤ 8, R ≥ 32 | MIC | ≥32 | R | S ≤ 8, R ≥ 32 | MIC | ≥32 | R |
| Ceftazidime | S ≤ 4, R ≥ 16 | MIC | ≤4 | S | S ≤ 4, R ≥ 16 | MIC | ≤4 | S |
| Ceftriaxone | S ≤ 1, R ≥ 4 | MIC | ≥64 | R | S ≤ 1, R ≥ 4 | MIC | ≥64 | R |
| Cefepime | S ≤ 2, R ≥ 16 | MIC | 16 | R | S ≤ 2, R ≥ 16 | MIC | 8 | SDD |
| Cefoxitin | S ≤ 8, R ≥ 32 | MIC | ≤8 | S | S ≤ 8, R ≥ 32 | MIC | ≤8 | S |
| Imipenem | S ≤ 1, R ≥ 4 | MIC | ≤1 | S | S ≤ 1, R ≥ 4 | MIC | ≤1 | S |
| Meropenem | S ≤ 1, R ≥ 4 | MIC | ≤1 | S | S ≤ 1, R ≥ 4 | MIC | ≤1 | S |
| Amikacin | S ≤ 16, R ≥ 64 | MIC | ≤4 | S | S ≤ 16, R ≥ 64 | MIC | ≤4 | S |
| Gentamicin | S ≤ 4, R ≥ 16 | MIC | ≤1 | S | S ≤ 4, R ≥ 16 | MIC | ≤1 | S |
| Ciprofloxacin | S ≤ 1, R ≥ 4 | MIC | 0.5 | S | S ≤ 1, R ≥ 4 | MIC | ≤0.06 | S |
| Levofloxacin | S ≤ 2, R ≥ 8 | MIC | 1 | S | S ≤ 2, R ≥ 8 | MIC | 1 | S |
| SXT | S ≤ 2, R ≥ 4 | MIC | ≥8/152 | S | S ≤ 2, R ≥ 4 | MIC | ≥8/152 | S |
| Polymyxin B | I ≤ 2 | MIC | ≤2 | I | I ≤ 2 | MIC | ≤2 | I |
| Nitrofurantoin | NA | NA | NA | NA | S ≤ 32, R ≥ 128 | MIC | ≤16 | S |
| Chloramphenicol | S ≤ 8, R ≥ 32 | MIC | ≤8 | S | NA | NA | NA | NA |
| Minocycline | S ≤ 4, R ≥ 16 | MIC | ≤4 | S | S ≤ 4, R ≥ 16 | MIC | ≤4 | S |

[a]NA, not available; R, resistant; S, susceptible; SDD, susceptible dose dependent.

## Genomic characterization of phage Henuyfy11N

Whole-genome sequencing revealed that the phage Henuyfy11N genome (GenBank accession number PV696613.1) is a circular double-stranded DNA molecule of a 41,103 bp with a G + C content of 50.87% (Fig. 3). A total of 54 ORFs were predicted, distributed across both strands (Table S2). Among these, 43 ORFs (79.63%) were assigned putative functions, while the remaining 11 (20.37%) were annotated as hypothetical proteins. The annotated ORFs were classified into six functional modules:

1. DNA, RNA, and nucleotide metabolism: 10 ORFs, including DNA helicase (ORF26, ORF33, and ORF34), DNA primase (ORF23), DNA polymerase (ORF28), nuclease superfamily protein (ORF30), ssDNA binding and annealing protein (ORF29), and a helix-turn-helix domain-containing protein (ORF33). ORF28 (DNA polymerase), ORF26 (DNA helicase), and ORF29 (ssDNA binding protein) showed high sequence identity (96.63%–100%) to corresponding genes in *Escherichia* phage BUCT789 (PQ999132.1).

2. Structural proteins, including two connector proteins (ORF4 and ORF6) and 12 tail-associated proteins, such as tail tube protein (ORF9), tail proteins (ORF3, ORF8, and ORF54), minor tail proteins (ORF14 and ORF15), tail length tape measure protein (ORF16), tail assembly chaperones (ORF18 and ORF19), tail fiber protein (ORF21), tailspike protein (ORF22), and a tail component (ORF7).

3. Head and packaging proteins, including large and small terminase subunits (ORF50 and ORF51), head scaffolding protein (ORF2), prohead protease (ORF5), and a 13.88 kDa late protein (ORF32).

4. Host lysis, containing endolysin (ORF44), lactocepin (ORF38), spanin (ORF1), and a class I holin-like protein (ORF43). ORF1, ORF43, and ORF44 exhibited 100% amino

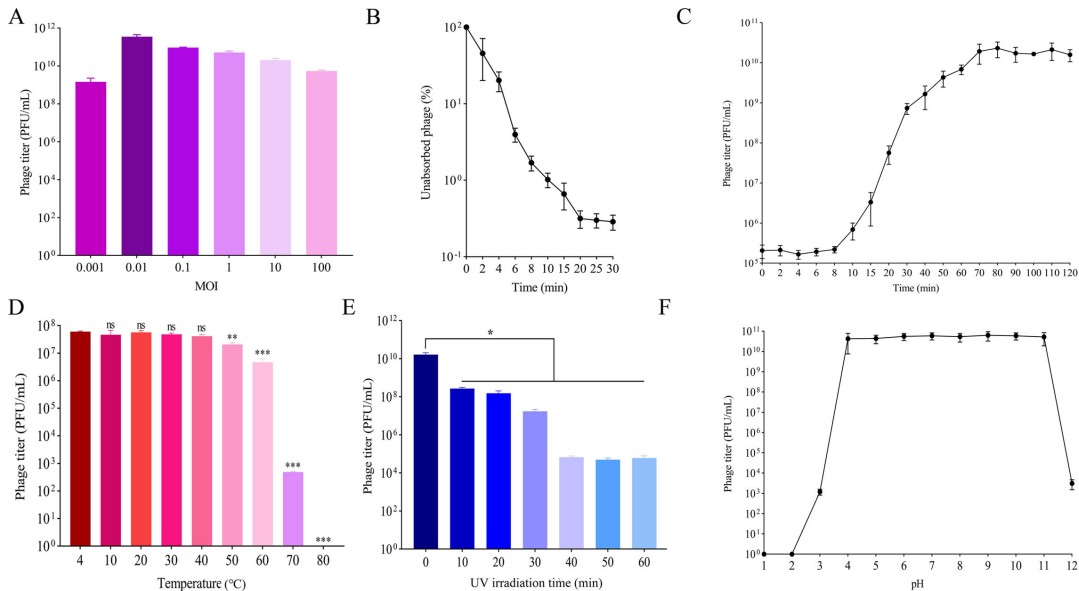

**FIG 2** Biological characteristics and stability of phage Henuyfy11N. (A) Optimal MOI of phage Henuyfy11N. (B) Adsorption rate of phage Henuyfy11N to its host, *E. coli* ECO11N. (C) One-step growth curve showing latent period, rise period, and burst size. (D) Thermal stability: survival rate of phage Henuyfy11N after incubation at different temperatures for 1 h. (E) UV stability: phage survival under UV irradiation (110 µW/cm²) over time. (F) pH stability: survival rate of phage Henuyfy11N after exposure to different pH values for 1 h. Data represent the mean ± 95% CI from three independent experiments ($n = 3$). Statistical significance was assessed using unpaired *t*-test (*$P < 0.05$, **$P < 0.01$, ***$P < 0.001$).

 acid identity to homologs in *Escherichia* phage BUCT789. The endolysin (ORF44) belongs to glycoside hydrolase family 19.

5. Moron, auxiliary metabolic genes, and host takeover, including calcineurin-like phosphoesterase (ORF12), glycoprotein precursor (ORF13), gamma-D-glutamyl-L-lysine dipeptidyl-peptidase (ORF20), cell division protein (ORF39), Nin protein (ORF45), and a zinc finger protein (ORF49).

6. Other phage-related proteins. Additionally, no tRNA, virulence, or antibiotic resistance genes were detected, confirming the lytic nature and suggesting the safety of phage Henuyfy11N for potential applications.

## Phylogenetic and comparative genomic analyses

A BLASTn search against the NCBI nucleotide database identified several phage genomes sharing sequence similarity with the Henuyfy11N, including *Escherichia* phages BUCT789 (PQ999132.1), phiSD1 (OP030970.1), SZH-1 (NC_073319.1), hz69 (ON556632.1), AV104 (OR352935.1), and EF2-1 (PQ571290.1). Synteny analysis indicated that all these phages encode DNA polymerase and DNA helicase, which are involved in DNA metabolism. Henuyfy11N encodes four lysis-associated proteins: endolysin, lactocepin, spanin, and class I holin-like protein. Comparative analysis revealed that endolysin is also present in BUCT789, phiSD1, EF2-1, and hz69; spanin is encoded by BUCT789, EF2-1, SZH-1, and phiSD1; and holin-like proteins are shared by SZH-1 and EF2-1, all showing high homology with corresponding genes in Henuyfy11N. Furthermore, tail fiber proteins are encoded by BUCT789, EF2-1, hz69, and AV104, while tailspike proteins are found in BUCT789, SZH-1, hz69, and AV104, also exhibiting high sequence homology with phage Henuyfy11N.

 VIRIDIC analysis revealed that phage Henuyfy11N shares 97.1% intergenomic similarity with *Escherichia* phage BUCT789, confirming that they belong to the same species. Additionally, Henuyfy11N showed 71.2%–77.3% similarity with phages phiSD1, SZH-1, hz69, AV104, and EF2-1, supporting their classification within the same genus according to standard VIRIDIC thresholds (70% for genus, 95% for species) (Fig. 4B).

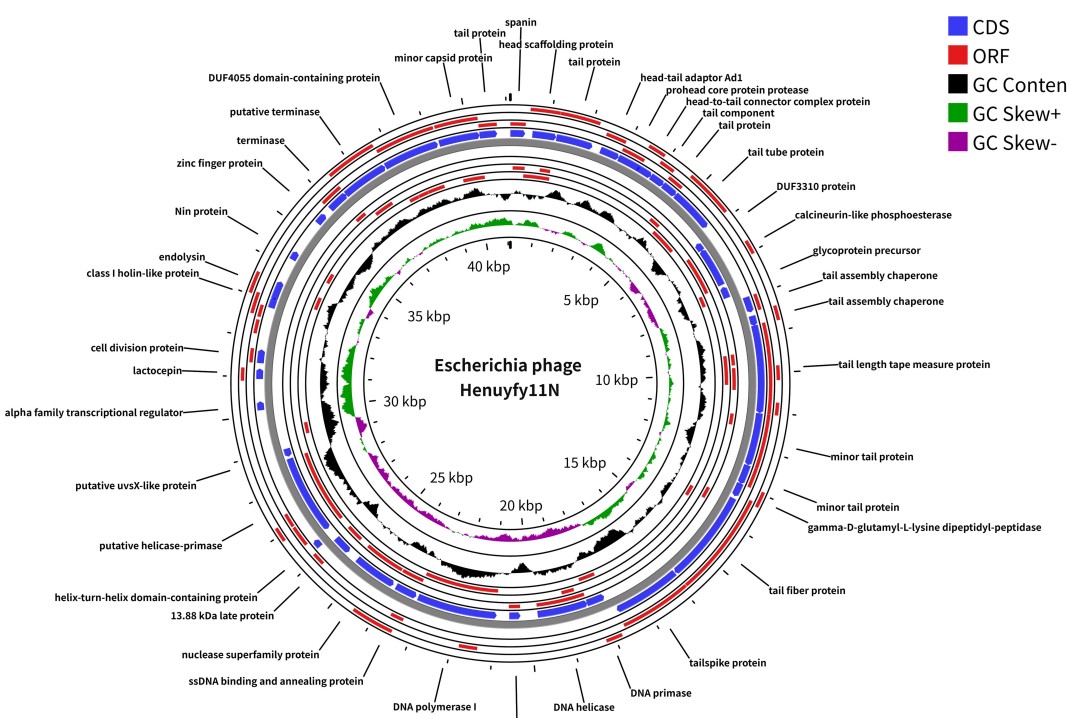

**FIG 3**  Genomic analysis of phage Henuyfy11N. Circular genome map of phage Henuyfy11N. From inside to outside: GC content (black), GC skew (green: positive; purple: negative), and open reading frames (ORFs) on forward (clockwise) and reverse (counterclockwise) strands.

Phylogenetic analysis based on whole-genome sequence similarity placed Henuyfy11N in close relation to *Escherichia* phage BUCT789, with which it shares 99.69% genomic identity (Fig. 5A). This classification was further corroborated by protein-level phylogenetic trees constructed from alignments of the DNA polymerase, terminase large subunit, and major capsid protein, all of which consistently grouped Henuyfy11N within clades of lytic *Escherichia* phages with high bootstrap support. Specifically, the DNA polymerase phylogeny indicated a close evolutionary relationship not only with *Escherichia* phage BUCT789 but also with *Escherichia* phage AV104 (Fig. 5C). Phylogenies derived from the major capsid protein and terminase large subunit reinforced a particularly strong affiliation with *Escherichia* phage BUCT789 (Fig. 5B and D). In summary, the whole-genome phylogenetic placement is consistent with the topologies obtained from the conserved protein phylogenies, confirming the reliable and coherent taxonomic assignment of phage Henuyfy11N.

## Bactericidal activity of Henuyfy11N *in vitro*

To evaluate the antibacterial capacity of phage Henuyfy11N, we assessed its ability to inhibit the growth of *E. coli* and its potential to suppress biofilm formation. The results demonstrated that phage Henuyfy11N significantly suppressed the growth of the host bacteria and delayed their entry into the stationary phase compared to the control group ($P < 0.05$, Fig. 6A). In the control group, the bacterial density ($OD_{600}$) increased steadily and reached the stationary phase after 210 min of cultivation. In contrast, all phage-treated groups exhibited effective growth inhibition throughout the first 8 h. Notably, no substantial differences in antibacterial efficacy were observed across the tested MOI values (0.01, 0.1, 1, and 10). The antibiofilm activity of phage Henuyfy11N was also evaluated. After 24 h of co-incubation at MOIs ranging from 0.01 to 10, a significant reduction in biofilm formation was observed, as quantified by $OD_{570}$ measurements (Fig. 6B). Furthermore, treatment with phage Henuyfy11N for 24 h at MOIs between 0.001

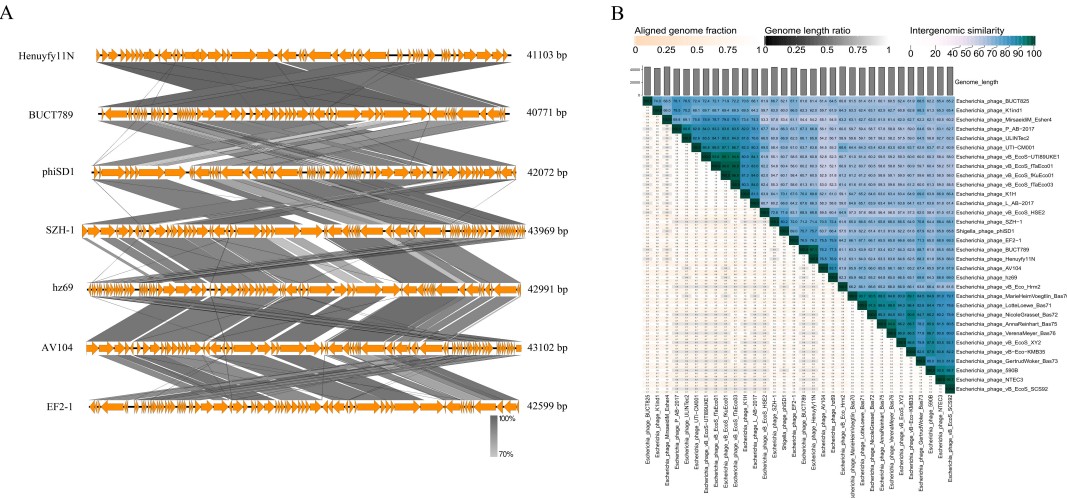

**FIG 4** Comparative genomic analysis of phage Henuyfy11N. (A) Whole-genome alignment of Henuyfy11N with related phages BUCT789, phiSD1, SZH-1, hz69, AV104, and EF2-1. (B) VIRIDIC-generated intergenomic similarity heatmap. The right color scale indicates percentage identity; the left axes show aligned fraction and genome length ratio.

and 10 resulted in a marked decrease in pre-formed biofilms compared to the untreated control (Fig. 6C).

## Therapeutic efficacy of Henuyfyfy11N in a mouse infection model

The antibacterial activity of phage Henuyfy11N was evaluated in a mouse model of systemic *E. coli* ECO11N infection. Mice were challenged with different bacterial doses to establish a lethal infection model. Compared to PBS-treated controls, infected mice exhibited clinical signs such as piloerection, lethargy, and reduced activity. All mice infected with high bacterial doses ($5.2 \times 10^{10}$, $2.6 \times 10^{10}$, and $1.3 \times 10^{10}$ CFU/mouse) succumbed to infection within 24 h (Fig. 7B). Those infected with $6.5 \times 10^9$ and $2.6 \times 10^9$ CFU/mouse reached 100% mortality by days 2 and 5, respectively (Fig. 7B). For phage therapy evaluation, two lethal challenge doses ($1.3 \times 10^{10}$ and $6.5 \times 10^9$ CFU/mouse) were selected. Mice administered phage alone at doses of $6.5 \times 10^{10}$ PFU/mouse and $1.3 \times 10^{11}$ PFU/mouse exhibited 100% survival, confirming the safety of Henuyfy11N (Fig. 7C and D). In mice infected with $1.3 \times 10^1$ CFU/mouse, phage treatment at MOIs of 0.1, 1, and 10 (corresponding to doses of $1.3 \times 10^9$, $1.3 \times 10^{10}$, and $1.3 \times 10^{11}$ PFU/mouse, respectively) significantly improved survival, with 20%–40% of animals surviving up to 7 days post-infection (Fig. 7C). Similarly, in the group infected with $6.5 \times 10^9$ CFU/mouse, phage treatment at the same MOIs (corresponding to doses of $6.5 \times 10^8$, $6.5 \times 10^9$, and $6.5 \times 10^{10}$ PFU/mouse, respectively) resulted in a survival rate of 60%–80% over the same period (Fig. 7D). The above results indicate that phage Henuyfy11N significantly improves the survival rate of mice infected with *E. coli*, demonstrating its potential for clinical therapeutic applications.

## DISCUSSION

In recent years, the extensive use of antimicrobial agents—particularly their inappropriate and incorrect application—has led to a continuous rise in drug resistance rates in *E. coli*. The problem of multidrug resistance has become increasingly severe, resulting in diminished efficacy, or even complete failure, of commonly used clinical antibiotics such as β-lactams and aminoglycosides (34–36). With the development of new antibacterial drugs facing a bottleneck, finding effective strategies to counteract bacterial resistance has become critically important. The objective of this study was to isolate and characterize novel bacteriophages capable of targeting ESBL-producing *E. coli*. These

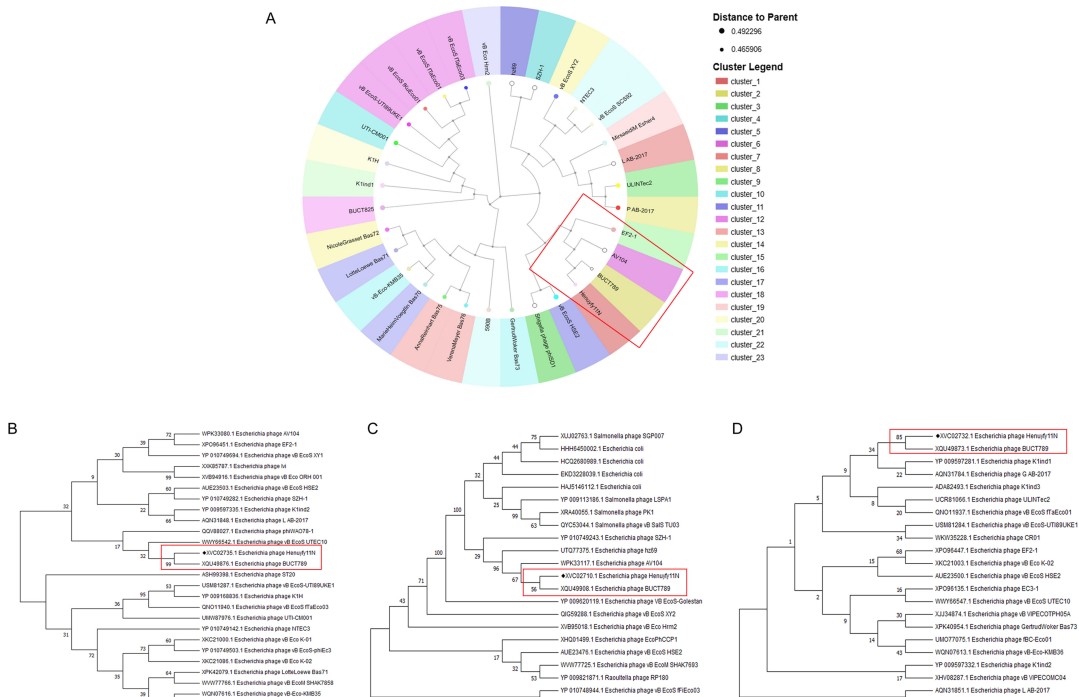

**FIG 5** Phylogenetic analysis of phage Henuyfy11N. Neighbor-joining phylogenetic trees constructed from (A) whole-genome sequences, (B) major capsid protein sequences, (C) DNA polymerase, and (D) terminase large subunit. The trees indicate close evolutionary relationships between Henuyfy11N and other phages. Bootstrap values are based on 1,000 replicates (MEGA 11).

bacteria are associated with a wide range of infections, including lower respiratory tract infections, bloodstream infections, urinary tract infections, and intra-abdominal infections. β-Lactam antibiotics have traditionally been the cornerstone of treatment for such infections. However, the escalating clinical use of these agents has led to a rising prevalence of ESBL-producing *Enterobacterales* (ESBL-E), contributing to a growing challenge of antimicrobial resistance that is now considered a critical public health issue worldwide. In 2024, the World Health Organization elevated ESBL-E to the second tier in its list of priority pathogens, underscoring the urgency of developing alternative therapeutics (37). Bacteriophages, with their capacity to specifically recognize and lyse bacterial hosts, have been recognized as potent biocontrol agents since their discovery over a century ago (38–40). While the development of a new antibiotic typically spans many years, the isolation and characterization of a novel phage can often be accomplished within days or weeks (41). Moreover, the bactericidal mechanisms of phages are distinct from those of antibiotics, which may reduce the risk of cross-resistance development. Phage-based treatments thus represent a promising strategy for mitigating infections and curbing the dissemination of antibiotic-resistant bacteria. In

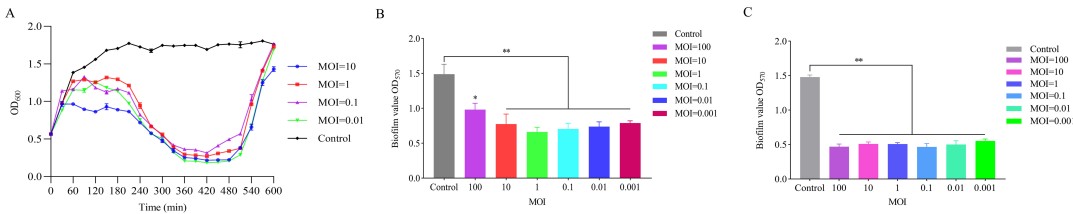

**FIG 6** *In vitro* antibacterial and antibiofilm activity of phage Henuyfy11N. (A) Growth inhibition of *E. coli* ECO11N by Henuyfy11N at MOIs of 0.01 to 10. Bacterial growth (OD$_{600}$) was monitored over 600 min. (B) Inhibitory effect of Henuyfy11N on biofilm formation by *E. coli* ECO11N. (C) Disruption of preformed *E. coli* ECO11N biofilms by phage Henuyfy11N. **$P < 0.01$.

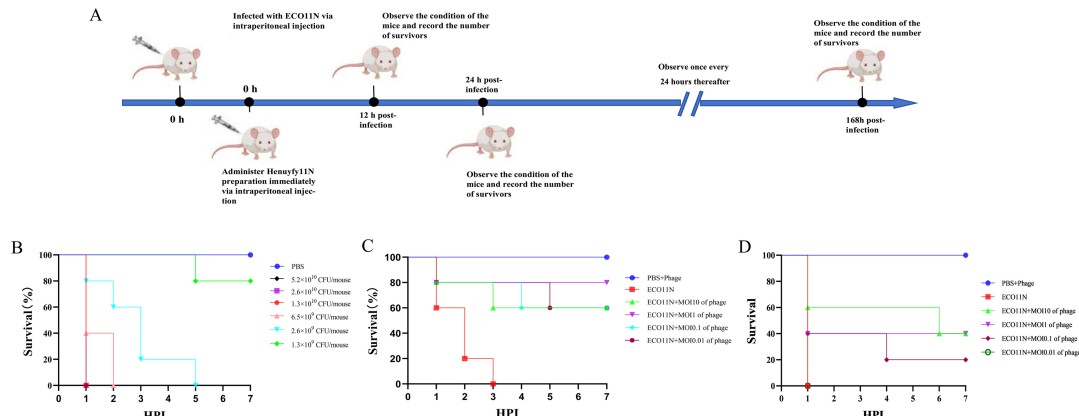

**FIG 7** Therapeutic efficacy of Henuyfy11N in a mouse infection model. (A) Schematic diagram of the experimental treatment protocol. (B) Survival curves of mice challenged with *E. coli* ECO11N or PBS (control). (C) Survival of mice infected with *E. coli* ECO11N ($6.5 \times 10^9$ CFU/mouse) and treated with phage Henuyfy11N or PBS. MOIs were calculated relative to the bacterial inoculum ($6.5 \times 10^9$ CFU/mouse): 0.01 ($6.5 \times 10^7$ PFU/mouse), 0.1 ($6.5 \times 10^8$ PFU/mouse), 1 ($6.5 \times 10^9$ PFU/mouse), and 10 ($6.5 \times 10^{10}$ PFU/mouse). (D) Survival of mice infected with *E. coli* ECO11N ($1.3 \times 10^{10}$ CFU/mouse) and treated with phage Henuyfy11N or PBS. MOIs were calculated relative to the bacterial inoculum ($1.3 \times 10^{10}$ CFU/mouse): 0.01 ($1.3 \times 10^8$ PFU/mouse), 0.1 ($1.3 \times 10^9$ PFU/mouse), 1 ($1.3 \times 10^{10}$ PFU/mouse), and 10 ($1.3 \times 10^{11}$ PFU/mouse).

this work, we isolated Henuyfy11N—a lytic phage targeting ESBL-producing *E. coli*—from wastewater samples collected at the First Affiliated Hospital of Henan University in Kaifeng. Its lytic activity and host specificity suggest potential therapeutic utility in combating infections caused by multidrug-resistant *E. coli*.

In this study, we performed a comprehensive analysis of the biological characteristics of the bacteriophage Henuyfy11N. This bacteriophage exhibited strict specificity towards *E. coli* and did not lyse *P. aeruginosa* or *A. baumannii*. This observation is consistent with certain narrow-host-range *E. coli* phages reported in the literature (42–44). These phages can only lyse specific *E. coli* strains during the infection process, with no impact on other bacterial species, indicating that Henuyfy11N has high host specificity, thereby enhancing its potential safety in therapeutic applications. In the stability assessment, Henuyfy11N maintained its activity within a temperature range of 4°C–60°C and a pH range of 4–11. It remained infectious after 30 min of ultraviolet exposure, demonstrating excellent physicochemical stability. This result is consistent with existing studies, highlighting the importance of phage stability in therapy, particularly considering the potential need for long-term survival under varying environmental conditions. Additionally, the one-step growth curve analysis of Henuyfy11N revealed a latency period of less than 10 min, reflecting its rapid infection cycle. This aligns with some phages reported in the literature, which also exhibit short latency periods accompanied by high lysis rates (45–48). However, compared to the longer latency periods and higher lysis numbers of some phages in other studies, Henuyfy11N, while having a short latency period, produced approximately 57.1 PFU/cell, indicating a moderate burst size. This observation may be related to its rapid infection cycle, suggesting that Henuyfy11N can efficiently spread in a short time, but its lysis amount is relatively moderate, possibly due to a balance between resource consumption during the lysis process and cellular metabolic mechanisms. The similarities and differences in these results may stem from the host specificities and evolutionary backgrounds of the phages.

The host range and growth characteristics of bacteriophages are often closely linked to their adaptability to hosts. The strict host specificity of Henuyfy11N may be attributed to its interaction mechanism with *E. coli*, while its lack of lytic activity against other bacteria could be related to its high affinity for surface receptors on *E. coli*. For example, a study by Gaborieau B in 2024, which utilized model predictions to analyze the infection relationship between bacteriophages and *E. coli*, indicates that host specificity is influenced not only by bacterial antibacteriophage mechanisms but

also by the adsorption factors of bacteriophages interacting with host cell surfaces (49). Furthermore, research conducted by Jia HJ and Loose M demonstrates that the host specificity of bacteriophages and their lytic activity against specific bacteria are generally closely associated with the binding affinity of bacteriophages to host cell surface receptors (50). We hypothesize that the differences in the latency period and lytic cycle of bacteriophage Henuyfy11N may be related to the bacteriophage's replication strategies and energy utilization efficiency within host cells. In summary, based on its characteristics, Henuyfy11N may have the potential to be a therapeutic bacteriophage, especially for *E. coli* infections.

The genomic annotation of phage Henuyfy11N reveals a highly functional array of genes essential for its lytic lifecycle, including endolysin (ORF44), a class I holin-like protein (ORF43), lactocepin, and spanin. These genes are characteristic of a lytic bacteriophage and are consistent with the observed cell-killing mechanisms of the phage. Host recognition and adsorption are crucial first steps in phage infection (51). Similar to many other bacteriophages, Henuyfy11N likely employs its tail fiber protein (ORF21) for host specificity, recognizing bacterial receptors on the cell surface, while the central spike protein (ORF22) facilitates membrane penetration, enabling the injection of the viral genome into the host cell (52–54). The late stages of the lytic cycle are marked by the action of endolysin (ORF44), which plays a pivotal role in degrading the bacterial cell wall, allowing the release of new phage progeny (55–57). Endolysins typically target the peptidoglycan layer, and their activity is often regulated by holin proteins that form membrane pores to allow endolysin access to its substrate (58, 59). In the case of phage Henuyfy11N, the holin-like protein ORF43 is likely involved in forming these pores, orchestrating a coordinated attack on the host cell. This collaborative action between tail fibers for host recognition and endolysins for cell lysis highlights a well-conserved mechanism in phage biology, observed in other lytic phages (60, 61). Bioinformatic analysis further supports the genetic safety of Henuyfy11N, as no known virulence factors, antibiotic resistance genes, or tRNA sequences were found in its genome. This is a key feature of many therapeutic phages, which need to be free from these genetic elements to ensure safety for potential applications in clinical and environmental settings (62–65). In contrast to some phages that harbor virulence genes or antibiotic resistance elements (66, 67), the absence of these factors in Henuyfy11N enhances its suitability for phage therapy, particularly for targeting antibiotic-resistant bacteria without the risk of contributing to resistance transmission. Moreover, the absence of tRNA genes in the Henuyfy11N genome aligns with the typical characteristics of strictly lytic phages, which do not rely on host machinery for protein synthesis during replication. This further reinforces the phage's efficient, self-contained lifecycle, independent of the host's translational apparatus. In comparison, temperate phages, which integrate their genome into the host's DNA, often carry tRNA genes or other genetic elements to facilitate their latent cycle (68, 69). Taken together, the genetic characterization of phage Henuyfy11N positions it as a possible candidate for therapeutic applications. Its well-defined lytic lifecycle, coupled with the absence of harmful genetic traits, ensures its potential for use in targeted bacteriophage therapy.

Biofilms represent a predominant mode of bacterial existence in both natural and clinical environments, conferring enhanced resistance to antibiotics, protection from phagocytosis, and increased tolerance to physical and environmental stresses (20, 70). Escherichia coli, like many other bacterial species, possesses a significant ability to form robust biofilms, complicating the treatment of infections associated with biofilm formation (71–73). Biofilm-related infections are notoriously difficult to treat due to their inherent resistance to antibiotic therapies, which are often ineffective against bacteria in this state. In recent years, bacteriophages have emerged as a promising alternative for combating biofilm-associated infections due to their ability to target and degrade biofilms. Previous studies have shown that phages can efficiently disrupt and eradicate biofilms in *vitro* and in animal models (25). This raises the potential of phage therapy as a targeted and effective strategy against biofilm-related infections, particularly in

cases where antibiotics fail. Our study demonstrated that phage Henuyfy11N exhibits strong antibiofilm activity against the *E. coli* ECO11N. After 24 h of treatment with phage Henuyfy11N, the majority of the pre-formed biofilm was eradicated, highlighting the phage's capacity to disrupt the biofilm matrix and eliminate bacterial cells embedded within it (74–76). This result aligns with previous research reporting the ability of bacteriophages to remove biofilms formed by *E. coli* and other pathogenic bacteria. In comparison to other bacteriophages, the narrow host specificity of Henuyfy11N enables targeted lysis of the primary pathogen embedded within the biofilm. This selectivity—combined with its self-amplifying activity and capacity to degrade the extracellular polymeric substance (EPS) matrix—synergistically enhances eradication efficacy relative to broad-spectrum antimicrobial agents, which often exhibit poor biofilm penetration and cause collateral damage to the resident microbial community (77–79). Some studies indicate that phage-based treatments may require higher doses or prolonged treatment periods to achieve similar results (80–82). However, Henuyfy11N demonstrates substantial biofilm removal within a relatively short time frame. This capacity to efficiently disrupt *E. coli* biofilms may be attributed to the unique properties of phage Henuyfy11N, including its rapid infection rate and the potential production of enzymes or proteins that degrade the extracellular matrix of the biofilm. The speed at which Henuyfy11N operates, combined with its effective biofilm penetration, provides it with a competitive advantage compared to other phages with slower infection cycles. In conclusion, our findings suggest that phage Henuyfy11N holds potential as a therapeutic agent for controlling biofilm-associated infections.

Bacteriophages are renowned for their ability to infect bacterial cells, replicate within them, and induce host cell lysis, leading to the release of progeny virions. This process can occur rapidly, often within hours, making phages an attractive therapeutic option for bacterial infections, particularly in cases where antibiotics fail (83, 84). In this study, phage Henuyfy11N demonstrated possible therapeutic efficacy in a mouse model of *E. coli* ECO11N bacteremia. Mice infected with *E. coli* ECO11N at a dose of $6.5 \times 10^9$ CFU/mouse exhibited 100% mortality within 72 h. However, treatment with phage Henuyfy11N at a dose of $10^9$ PFU/mouse resulted in an 80% survival rate over 7 days, highlighting the potential of this phage as a possible therapeutic agent for combating *E. coli* infections. These results are consistent with previous studies demonstrating the potential therapeutic efficacy of bacteriophages in mouse models of bacteremia (85, 86). Although phage Henuyfy11N exhibited possible therapeutic efficacy, it did not achieve a 100% survival rate. This outcome is not unexpected, as phage resistance can emerge in *vivo*, complicating the efficacy of phage therapy. Phage-resistant variants, often arising from mutations in bacterial surface receptors or the production of antiphage enzymes, represent a well-documented challenge in the clinical application of phage therapy (87, 88). The lack of complete survival in this study may also be attributed to the complexities of the in *vivo* environment, where factors such as immune response, tissue distribution, and phage pharmacokinetics significantly influence therapeutic outcomes. Previous research has indicated that the success of phage therapy can be affected by factors such as phage dose, frequency of administration, and the presence of other bacterial strains or biofilms that may shield bacteria from phage attack (89, 90). Furthermore, the emergence of phage-resistant bacterial populations remains one of the most significant obstacles to the widespread use of phages as therapeutics. While phage Henuyfy11N demonstrated impressive efficacy in this study, further research is necessary to evaluate its ability to overcome resistance mechanisms, potentially through strategies such as phage cocktails or genetic modifications to enhance phage activity (91, 92). In terms of safety, phage Henuyfy11N exhibited a favorable profile, with a compact genome of 41,103 bp and no detectable antimicrobial resistance or virulence genes. This aspect is critical for the development of phage-based therapies, as the potential for horizontal gene transfer between phages and bacteria could lead to the dissemination of undesirable traits, such as antibiotic resistance (93, 94). The absence of such genes in phage Henuyfy11N underscores its potential as a safe and effective biological agent for

therapeutic use. Additionally, its efficacy against ESBL-producing *E. coli* strains highlights its promise in treating infections caused by multidrug-resistant bacteria, which continue to be a growing concern in clinical settings. Compared to other phages employed in similar therapeutic applications, phage Henuyfy11N distinguishes itself due to its specific activity against *E. coli* and its robust performance in an in *vivo* model (95–98).

Despite Henuyfy11N demonstrating potent inhibitory effects against ESBL-producing *E. coli* in both in *vitro* and in *vivo* models, with no known virulence or resistance genes identified and a favorable safety profile and clinical application prospects, this study has several limitations. The evidence is primarily based on acute, relatively simple infection models, which may not be easily extrapolated to complex clinical situations involving device-associated infections or immunocompromised patients. Additionally, the scale and phylogenetic diversity of the strain library are limited, and the applicability to carbapenemase-producing strains and colistin-resistant strains warrants cautious interpretation. Host range determinants and receptor mechanisms have not been systematically elucidated, and longitudinal data on phage resistance occurrence rates, molecular basis, and fitness costs are lacking. Furthermore, there is insufficient quantitative evidence regarding pharmacokinetics, pharmacodynamics, tissue penetration capability, optimal dosage, administration routes, and treatment duration. The immunogenicity and safety margins for various administration routes and dosages require further strengthening. Concurrently, improvements are needed in production scale-up, endotoxin control, long-term stability, and companion diagnostics, including rapid sensitivity screening, in *vivo* kinetics, and neutralizing antibody monitoring.

Future studies should conduct multicenter, multilineage large-scale strain evaluations combined with mechanistic studies to analyze receptors and resistance trajectories, aiming to optimize cocktail therapy and antibiotic combinations. Establishing PK/PD models for different infection sites will guide the optimization of dosage and administration routes while systematically validating efficacy against device-associated infections. It is also essential to enhance GMP-grade processes and key quality standards, reduce endotoxin burden, and improve formulation stability. Developing rapid companion diagnostics and therapeutic monitoring tools will support personalized and adaptive dosing. Finally, advancing prospective, adaptive-design early clinical trials in high medical need indications, such as recurrent urinary tract infections and device-associated infections, should utilize composite endpoints that include clinical outcomes, microbiological clearance, and safety. Overall, Henuyfy11N, with its high specificity for *E. coli*, potential efficacy against ESBL-producing *E. coli*, and favorable genomic safety profile, already lays the foundation to become a precision antimicrobial biological agent.

## ACKNOWLEDGMENTS

We thank the First Affiliated Hospital of Henan University for the donation of strains tested in our host range analysis.

The present study is supported by the provinces and ministries jointly building key projects of the Henan Provincial Health Commission (SBGJ202402084 and SBGJ202302089), the platform construction fund of the Henan Province Engineering Technology Research Center of Rapid-Accuracy Medical Diagnostics (30389), and the Henan Provincial Program for Medical Science and Technology Development (LHGJ20240383).

L.W. designed the study. X.Z. and J.H. isolated the phage. X.Z. and D.Q. performed the experiments. Z.L. analyzed the genomic data. Q.L. wrote the manuscript. All authors approved the final version.

## AUTHOR AFFILIATIONS

[1]Henan Province Engineering Technology Research Center of Rapid-Accuracy Medical Diagnostics, The First Affiliated Hospital of Henan University, Henan University, Kaifeng, China

²Department of Clinical Laboratory, The First Affiliated Hospital of Henan University, Henan University, Kaifeng, China
³Kaifeng Engineering Technology Research Center for *In Vitro* Diagnostics, The First Affiliated Hospital of Henan University, Henan University, Kaifeng, China

## AUTHOR ORCIDs

Qiming Li  http://orcid.org/0000-0002-6916-3954
Li Wang  http://orcid.org/0009-0001-1822-5887

## FUNDING

| Funder | Grant(s) | Author(s) |
| --- | --- | --- |
| Health Commission of Henan Province | SBGJ202402084, SBGJ202302089 | Li Wang |
| Science and Technology Department of Henan Province | 30389 | Li Wang |
| Health Commission of Henan Province | LHGJ20240383 | Xinwei Zhang |

## AUTHOR CONTRIBUTIONS

Xinwei Zhang, Data curation, Investigation, Writing – original draft | Jiaqi He, Investigation, Software | Dongliang Qiao, Software, Visualization | Qiming Li, Methodology, Supervision | Zhigang Liu, Writing – original draft, Writing – review and editing | Li Wang, Funding acquisition, Project administration

## DATA AVAILABILITY

The genome and annotation of *Escherichia* phage Henuyfy11N have been submitted to the GenBank database with accession number PV696613.1.

## ETHICS APPROVAL

The study was approved by the First Affiliated Hospital of Henan University Ethics Committee, and ethics number HUSOM2026-017 was assigned to it. Informed consent was acquired for clinical and biological information. All research conducted in this study complies with the Declaration of Helsinki.

## ADDITIONAL FILES

The following material is available online.

### Supplemental Material

**Tables S1 and S2, and Figure S1 (Spectrum03253-25-s0001.docx).** Tables S1: Strains used for phage Henuyfy11N host range analysis. Tables S2: The open reading frames (ORFs) analysis of phage Henuyfy11N. Fig. S1: Taxonomic prediction of phage Henuyfy11N using taxMyPhage v3.3.6.

### Open Peer Review

**PEER REVIEW HISTORY (review-history.pdf).** An accounting of the reviewer comments and feedback.

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
