## [Reviewer comments · Microbiology Spectrum]

Microbiology Spectrum

Characterization of Novel Phage Henufy11N: A Potential Therapeutic Agent against Extended-Spectrum β -Lactamase (ESBL)-Producing *Escherichia coli*

Li Wang, Xinwei zhang, Jiaqi He, qiming li, Dongliang Qiao, and Zhigang Liu

Corresponding Author(s): Li Wang, The First Affiliated Hospital of Henan University

Review Timeline:

Submission Date:	October 9, 2025
Editorial Decision:	November 20, 2025
Revision Received:	January 10, 2026
Editorial Decision:	February 9, 2026
Revision Received:	March 3, 2026
Accepted:	March 20, 2026

Editor: Sadjia Bekal

Reviewer(s): Disclosure of reviewer identity is with reference to reviewer comments included in decision letter(s). The following individuals involved in review of your submission have agreed to reveal their identity: Saisubramanian Nagarajan (Reviewer #2)

Transaction Report:

DOI: <https://doi.org/10.1128/spectrum.03253-25>

Re: Spectrum03253-25 (Characterization of Novel Phage Henufy11N: A Potential Therapeutic Agent against Extended-Spectrum β -Lactamase (ESBL)-Producing *Escherichia coli*)

Dear Dr. Li Wang:

Thank you for the privilege of reviewing your work. Below you will find my comments, instructions from the Spectrum editorial office, and the reviewer comments.

Revision Guidelines

Sincerely,
Sadjia Bekal
Editor
Microbiology Spectrum

Reviewer #1 (Comments for the Author):

Dear Authors

The Following a detailed review, the following points require attention and revision to enhance the scientific rigor, clarity, and overall quality of the paper. The feedback is organized by section for your convenience.

Materials and Methods Section

Completing Details: Please ensure the full name, manufacturer, city, and country are provided for all materials, kits, and equipment used.

Referencing Protocols: For all experiments performed (e.g., Phage adsorption assay, EOP and Burst Size calculations), the source (reference) and specific protocol must be cited.

Clarifying Calculations and Rationale:

The method and mathematical formula used to calculate the Burst size (line 18) and the Efficiency of Plating (EOP) (lines 118-119) must be transparently described in the text.

The scientific justification for selecting an MOI of 0.01 for the phage adsorption assay (line 129) needs to be provided.

The rationale behind using "phage progeny production" instead of "reduction in bacterial growth" to determine the optimal MOI (line 141) requires explanation.

Solution Composition: The exact components and concentrations of the "phage precipitation solution" (lines 197-198) must be clearly stated in the text or methods section.

Sequencing Platform: The manufacturer, model, and country of origin of the sequencing device used (line 217) should be specified.

Bacterial Nomenclature: Throughout the text, particularly in line 109, bacterial names must be corrected and standardized according to international conventions (i.e., genus and species names in italics).

Minor Corrections:

The origin of the phage (lines 86-88) must be clarified (obtained from another institute or isolated in this study).

The formatting 12,000× g (line 91) should be corrected to 12,000 ×g (with a space before the multiplication sign).

Results Section

Defining Concepts: The specialized term Synteny (line 16) requires a brief, practical definition within the text.

Statistical Analyses: It is necessary to specify the name of the statistical software, the exact tests used, and the significance level (P-value) applied for data analysis in various sections (e.g., biological characteristics and stability tests of the phage).

Explaining Abbreviations: The abbreviation SDD in Table 1 must be fully explained in the table footnote or the methods section.

Figure Revisions and Explanations:

In Figure 2-C, given the curve's shape, the method for calculating the Latent Period = 10 minutes needs more detailed explanation.

A clear title or description should be added to Figure 2-E to indicate which parameter the graph represents.

Please verify the correct spelling of pH on the axis label in Figure 2-F.

The figure referenced in lines 313-314 is missing and must be added.

The exponential growth of bacteria in all treatments in Figure 6-A after 480 minutes requires scientific interpretation and justification to explain why the phage could not effectively control the growth.

Discussion & Conclusion Section

This section requires substantial rewriting and strengthening. A robust scientific discussion should include:

A detailed and analytical comparison of the findings from this study with results from similar published research.

An interpretation of the reasons behind the observed similarities and differences with previous work.

A statement of the limitations of the present study.

A strong final conclusion and suggestions for future research directions.

Ethical Compliance

Missing Ethical Approval: The manuscript currently lacks a statement regarding ethical approval for animal experiments. If any part of this research involved laboratory animals, it is mandatory to provide the name of the ethics committee that approved the study and the associated ethical approval code. This statement must be included in the Materials and Methods section.

Technical Corrections and References

Referencing:

The source for the BLAST analysis (line 223) must be added.

The phage taxonomic classification should be updated according to the latest ICTV changes (citing the 2025 article provided <https://doi.org/10.1099/jgv.0.002111>) (lines 264-266).

Critical Errors:

The family name Muscoviridae (line 17) is incorrect and must be corrected by consulting the official ICTV database.

The Accession Number reported in line 225 belongs to a coronavirus protein and is erroneous. It must be replaced with the correct Accession Number corresponding to the phage genome sequenced in this study.

Reviewer #2 (Comments for the Author):

The research manuscript describes the isolation, characterization, and therapeutic evaluation of bacteriophage Henufy11N targeting ESBL-producing *E. coli*. Although the study tackles a significant clinical issue, the manuscript needs to be clarified on methodological issues, problems with data interpretation.

MAJOR CONCERNS

1. The authors state that phage Henufy11N has been classified as a member of the family "Muscoviridae" according to ICTV 2022 guidelines. However, no such family exists in the current ICTV taxonomy database. Please clarify and justify the taxonomic assignment of this phage. Provide evidence from phylogenetic analysis and cite the correct ICTV-recognized family.

Materials and Methods

2. In host specificity, authors mentioned using PBS for diluting the phages. Whether authors use PBS alone or was it supplemented with $MgCl_2$ and $CaCl_2$? How might phage stability be affected by using standard PBS without divalent cations? Pls explain.

3. In pH stability assays, authors mentioned using phage solution without specifying its composition. Please specify what is phage solution (e.g., buffer type, ionic strength, presence of stabilizers)?

4. In One-step growth curve assay, The authors report a high centrifugation speed at $12,000 \times g$ to pellet bacterial cells, relative to commonly used $4,000-6,000 \times g$ in bacteriophage one-step growth experiments. Is there a specific reason for using higher speeds? Will it not affect viability. I am not sure if this is connected but there is hardly any latent phase in the one step growth curve results

5. Usually SM buffer or phage buffer is preferred for phage isolation. Did the authors test both buffers before opting for PBS?

Results

6. In the Morphological characterization of phage Henufy11N, Figure 1C and D appears slightly different are they one or 2 different phages?

7. The phage shows lytic activity against only two *E. coli* strains out of 150 tested (1.3%), and only a single phage was isolated. Given that bacteria can rapidly develop phage resistance, the therapeutic relevance of narrow-host-range phage is a bit limited.

8. In the biological characteristics section, line 288, authors claim a latent period of 10 mins. However Figure 2C, there is a rise in phage titre from the 0th time point, making the latent period is unclear. Please clarify the reason for the absence of latent phase.

9. In pH stability graph (Figure 2F), phages display unusually high activity at pH 4. Provide an explanation or mechanistic basis for this high activity at acidic pH, How many times the experiments were repeated was it biological/technical replicates.

10. In the Figure 2B, the number of time points is limited, and the adsorption rate does not reach plateau state. How can authors claim that this is fully adsorbed?

11. In the restriction enzyme analysis, authors are claiming that 2 *Xba*I recognition sites, 2 *Spe*I is there. But the gel image Figure 3A, shows a single band similar in length to the control. Did the digestion happen?

12. Figure 6A clearly shows bacterial regrowth and resistance emergence after 10 hours of phage treatment. Given the emergence of resistance in short time span how can this phage be therapeutically effective?

13. As the phage resistant bacterial population arose within 10 hrs, yet the animal study shows improved survival over several days. Will resistant mutant not arise in animal model of infection? If the rate of resistant mutant generation varies between in vitro and in vivo condition, the authors should provide appropriate literature support and explain the discrepancy between in vitro and in vivo results.

14. If the authors have already determined optimal MOI in vitro why did the authors preferred to test different MOI in vivo?

- Upload point-by-point responses to the issues raised by the reviewers in a file named "Response to Reviewers," NOT in your cover letter.
- Upload a compare copy of the manuscript (without figures) as a "Marked-Up Manuscript" file.
- Upload a clean .DOC/.DOCX version of the revised manuscript and remove the previous version.
- Each figure must be uploaded as a separate, editable, high-resolution file (TIFF or EPS preferred), and any multipanel figures must be assembled into one file.
- Any supplemental material intended for posting by ASM should be uploaded with their legends separate from the main manuscript. You can combine all supplemental material into one file (preferred) or split it into a maximum of 10 files with all associated legends included.

Thanks to the editorial team for the reminder, we made the changes exactly as requested.

Reviewer #1 (Comments for the Author):

Dear Authors

The Following a detailed review, the following points require attention and revision to enhance the scientific rigor, clarity, and overall quality of the paper. The feedback is organized by section for your convenience.

Materials and Methods Section

Completing Details: Please ensure the full name, manufacturer, city, and country are provided for all materials, kits, and equipment used.

Thank you sincerely for your constructive and precise suggestions! we have supplemented the missing content mentioned above.

Referencing Protocols: For all experiments performed (e.g., Phage adsorption assay, EOP and Burst Size calculations), the source (reference) and specific protocol must be cited.

Thank you sincerely for your constructive and precise suggestions! we have cited the correct references.

Clarifying Calculations and Rationale:

The method and mathematical formula used to calculate the Burst size (line 18) and the Efficiency of Plating (EOP) (lines 118-119) must be transparently described in the text.

Thank you sincerely for your constructive and precise suggestions! we have supplemented the missing content mentioned above.

The adsorption rate constant for phage Henufy11N was calculated using the equation below

$$k = -\ln(P/P_0)/B_t(1),$$

k is the adsorption rate constant (mL/min), P is the free phage concentration per mL, P_0 is the initial phage concentration, B is the initial bacterial density, and t is the time (min).

Burst Size = (Total Progeny at final plateau) / (Number of infected cells at initial plateau) (2).

EOP = (average PFU on target bacteria) / (average PFU on host bacteria) (3,4).

1. Gomaa SE, Wang Y, Li J, Cao H, Shen J, Dong X, Liu Y, Chen N, Li Q, Yan Z et al: **Phage Henu12-resistant mutant derives fitness trade-offs in *Shigella dysenteriae***. Microbiology spectrum 2025: e0115025.
2. Kropinski AM: **Practical Advice on the One-Step Growth Curve**. *Methods in molecular biology (Clifton, NJ)* 2018, **1681**:41-47.
3. Hall-Stoodley L, Stoodley P: **Evolving concepts in biofilm infections**. Cellular microbiology 2009, 11(7):1034-1043.

4. Kutter E: **Phage host range and efficiency of plating.** *Methods in molecular biology (Clifton, NJ)* 2009, **501**:141-149.

The scientific justification for selecting an MOI of 0.01 for the phage adsorption assay (line 129) needs to be provided.

In adsorption experiments, our objective is to accurately measure the proportion of phages adsorbed to host cells over a specified duration. An improper selection of the multiplicity of infection (MOI) can introduce various errors, leading to an inaccurate representation of the phage's adsorption capacity.

(1) It is essential to ensure that the assumption of "single adsorption" is valid to avoid re-adsorption. During centrifugation to remove the supernatant and terminate adsorption, cells bound to multiple phages may retain one phage while others detach into the supernatant during the resuspension process. This situation can result in an underestimation of the total number of phages that actually underwent adsorption, as the detached phages are counted as unadsorbed.

(2) The "saturation" effect on the surface of host cells must be avoided. When utilizing an excessively high MOI, phage particles can rapidly occupy all available receptor sites, causing the adsorption curve to plateau (100% adsorption) in a very short time. This phenomenon complicates the observation of the dynamics of the phage adsorption process.

(3) It is crucial to minimize the "self-interference" effect. When multiple phages approach a cell simultaneously, they may physically obstruct each other.

In summary, operating under the optimal MOI (0.01) conditions simulates an infinitely diluted environment that allows the adsorption process to follow first-order kinetics. This approach facilitates the accurate measurement of the adsorption rate constant while avoiding various experimental artifacts (5,6). We have extended the time points to determine whether the plateau has been reached, thereby ensuring a comprehensive capture of the adsorption kinetics, as illustrated in the subsequent figure.

5. Hyman, P., Abedon, S. T. Practical methods for determining phage growth parameters. In: **Clokier, M. R. J., Kropinski, A. M. (eds) Bacteriophages: methods and protocols, volume 1: isolation, characterization, and interactions.** *Methods in Molecular Biology*, vol 501. Humana Press; 2009.
6. Yan T, Wang Q, Ma C, Teng X, Gong Z, Chu W, Zhou Q, Liu Z: **Phage vB_Kpn_HF0522: Isolation, Characterization, and Therapeutic Potential in Combatting K1 *Klebsiella pneumoniae* Infections.** *Infection and drug resistance* 2025, **18**:803-818.

The rationale behind using "phage progeny production" instead of "reduction in bacterial growth" to determine the optimal MOI (line 141) requires explanation.

Thank you for your thorough review of the article and your highly professional feedback. In phage therapy, the hallmark of success lies not only in the initial bactericidal effect but also in the phage's ability to persist at the infection site, self-amplify, and ultimately clear the infection. This dynamic biological process relies on self-replication.

(1) A reduction in bacterial growth reflects static killing rather than dynamic proliferation. While it indicates the extent of bacterial clearance at a specific time point, it does not provide insights into the phage's replication status. It fails to distinguish the underlying causes of "bacterial death." The decrease in bacterial numbers may result from successful phage replication and lysis, which is desired. Alternatively, it could stem from "external killing" due to multiple infections, where a bacterium is simultaneously infected by several phages, leading to rapid lysis but potentially insufficient resources for each phage to produce a substantial number of progenies. Additionally, other non-replicative factors, such as toxins or the bacterial stress response, may contribute to this outcome. Consequently, this could lead to an overestimation of the multiplicity of infection (MOI); a high MOI can indeed kill bacteria rapidly, as nearly every bacterium is infected simultaneously. However, this represents a form of "resource waste," where numerous phage particles attach to a limited number of bacteria, resulting in many phages missing the opportunity to complete their replication cycle and yielding a very low overall progeny output. Furthermore, the reduction in bacterial growth overlooks the characteristic "one-step growth" of phages, which includes an inherent latent period and burst size. Simply observing bacterial reduction misses critical information regarding the completion of the replication cycle.

(2) Progeny yield directly reflects the phage's capacity to complete the entire infection-replication-assembly-release cycle. A high progeny yield indicates that the infection conditions, including MOI, are optimal, allowing the phage to efficiently utilize host resources for maximum amplification.

In summary, selecting "phage progeny yield" as the gold standard for determining the optimal MOI is grounded in an understanding of the biological essence of phages and the fundamental objectives of phage therapy. For phage therapy aimed at eradicating infections, ensuring effective phage proliferation within the host is significantly more crucial than the initial bactericidal rate (7,8.)

7. Kutter E. Phage host range and efficiency of plating. In: **Clokie MR, Kropinski AM, editors. Bacteriophages: Methods and Protocols, Volume 1: Isolation, Characterization, and Interactions.** Totowa, NJ: Humana Press; 2009. p. 141–149. (Methods in Molecular Biology; 501).
8. Cao S, Yang W, Zhu X, Liu C, Lu J, Si Z, Pei L, Zhang L, Hu W, Li Y et al: **Isolation and identification of the broad-spectrum high-efficiency phage vB_SalP_LDW16 and its therapeutic application in chickens.** BMC veterinary research 2022, 18(1):386.

Solution Composition: The exact components and concentrations of the "phage precipitation solution" (lines 197-198) must be clearly stated in the text or methods section.

Thank you sincerely for your constructive and precise suggestions. We have addressed the missing content you mentioned and included it in the Materials and Methods section. The primary components of the phage precipitation solution are 20% polyethylene glycol and 2.5 M

sodium chloride. To prepare the solution, dissolve 200 grams of polyethylene glycol in approximately 800 mL of deionized water. Next, add 146.1 grams of sodium chloride, stirring until it is completely dissolved, and then adjust the total volume to 1 liter with deionized water (9).

9. Li Q, Li J, Zhao Y, Guo S, Liu M, Shi X, Wang L, Liu Z, Teng T: **Characterization and genomics of phage Henu2_3 against K1 *Klebsiella pneumoniae* and its efficacy in animal models**. *AMB Express* 2025, 15(1):112.

Sequencing Platform: The manufacturer, model, and country of origin of the sequencing device used (line 217) should be specified.

Thank you sincerely for your constructive and precise suggestions! we have supplemented the missing content mentioned above.

Bacterial Nomenclature: Throughout the text, particularly in line 109, bacterial names must be corrected and standardized according to international conventions (i.e., genus and species names in italics).

Thank you sincerely for your constructive and precise suggestions! we have corrected the mistakes mentioned above.

Minor Corrections:

The origin of the phage (lines 86-88) must be clarified (obtained from another institute or isolated in this study).

Thank you sincerely for your constructive and precise suggestions! we have supplemented the missing content mentioned above.

The formatting $12,000 \times g$ (line 91) should be corrected to $12,000 \times g$ (with a space before the multiplication sign).

Thank you sincerely for your constructive and precise suggestions! we have corrected the mistakes mentioned above.

Results Section

Defining Concepts: The specialized term Synteny (line 16) requires a brief, practical definition within the text.

Thank you sincerely for your constructive and precise suggestions! we have supplemented the missing content mentioned above.

Statistical Analyses: It is necessary to specify the name of the statistical software, the exact tests used, and the significance level (P-value) applied for data analysis in various sections (e.g., biological characteristics and stability tests of the phage).

Thank you sincerely for your constructive and precise suggestions! we have supplemented the missing content mentioned above.

Explaining Abbreviations: The abbreviation SDD in Table 1 must be fully explained in the table footnote or the methods section.

Thank you sincerely for your constructive and precise suggestions! we have supplemented the missing content mentioned above.

Figure Revisions and Explanations:

In Figure 2-C, given the curve's shape, the method for calculating the Latent Period = 10 min needs more detailed explanation.

Thank you for your valuable suggestions. The latency period of bacteriophages is not determined using a single formula; rather, it is directly observed and determined through a one-step growth

experiment that analyzes the dynamic changes in the number of infectious bacteriophages, as depicted in the growth curve. Specifically, it is directly read from the one-step growth curve by measuring the time elapsed from the initial time point to the moment when the curve first begins to rise significantly. Following the reviewer's suggestion, we increased the number of time points within the first 20 min to more accurately observe the changes in the latent period. The new data reveal that the phage has a relatively short latent period of approximately 8 min, and this result has been updated in the figure. We have revised Figure 2C accordingly and made the necessary corrections to the latent period in the biological characteristics section.

A clear title or description should be added to Figure 2-E to indicate which parameter the graph represents.

Thank you sincerely for your constructive and precise suggestions! we have supplemented the missing content mentioned above.

Please verify the correct spelling of pH on the axis label in Figure 2-F.

Thank you sincerely for your constructive and precise suggestions! we have corrected the mistakes mentioned above.

The figure referenced in lines 313-314 is missing and must be added.

Thank you sincerely for your constructive and precise suggestions! we have supplemented the missing content mentioned above. We placed it at the position shown in Figure 2C.

The exponential growth of bacteria in all treatments in Figure 6-A after 480 min requires scientific interpretation and justification to explain why the phage could not effectively control the growth.

Thank you for your thorough review of the article and your highly professional feedback. As the "predation" pressure from bacteriophages persists, bacterial populations will evolve under intense natural selection, leading to the following resistance mechanisms that contribute to the "failure" of bacteriophages:

(1) Bacteria develop resistant mutations, which are the primary mechanism of resistance. Through spontaneous mutations, bacteria undergo minor alterations in their genomes, thereby acquiring

resistance to specific bacteriophages. These mutations primarily occur in structures associated with the adsorption and invasion of bacteriophages. Additionally, bacterial restriction-modification systems, the CRISPR-Cas system, and abortive infection systems also play significant roles.

(2) Once a bacterial mutant exhibiting any of the aforementioned resistance mechanisms emerges, it gains a substantial growth advantage in environments where bacteriophages have eliminated all sensitive competitors. Such mutants can fully exploit the available nutrients and space for rapid proliferation, ultimately becoming the dominant strain within the population.

Discussion & Conclusion Section

This section requires substantial rewriting and strengthening. A robust scientific discussion should include:

A detailed and analytical comparison of the findings from this study with results from similar published research.

An interpretation of the reasons behind the observed similarities and differences with previous work.

A statement of the limitations of the present study.

A strong final conclusion and suggestions for future research directions.

We sincerely appreciate the reviewer's comments, which have significantly enhanced the clarity, rigor, and interpretability of the discussion section. We have completely revised the Discussion and addressed all requested elements.

Ethical Compliance

Missing Ethical Approval: The manuscript currently lacks a statement regarding ethical approval for animal experiments. If any part of this research involved laboratory animals, it is mandatory to provide the name of the ethics committee that approved the study and the associated ethical approval code. This statement must be included in the Materials and Methods section.

Thank you sincerely for your constructive and precise suggestions! we have supplemented the missing content mentioned above. We placed it in the Materials and Methods section.

Technical Corrections and References

Referencing:

The source for the BLAST analysis (line 223) must be added.

Thank you sincerely for your constructive and precise suggestions! we have supplemented the missing content mentioned above.

The phage taxonomic classification should be updated according to the latest ICTV changes (citing the 2025 article provided <https://doi.org/10.1099/jgv.0.002111>) (lines 264-266).

Thank you sincerely for your constructive and precise suggestions! We have made revisions in accordance with your suggestions.

Critical Errors:

The family name Muscoviridae (line 17) is incorrect and must be corrected by consulting the official ICTV database.

Thank you sincerely for your constructive and precise suggestions! we have corrected the mistakes mentioned above. The phage Henuyfy11N is classified within the same genus as *Escherichia* phage SZH-1 (NC_073319.1). SZH-1 is categorized under the *Kagunavirus* genus at the genus level in various phage genome databases, including Millard Lab Phage Genomes and VirusHostDB. However, its family affiliation has not been formally confirmed by the ICTV. Consequently, its family status is categorized as unclassified or incertae sedis.

The Accession Number reported in line 225 belongs to a coronavirus protein and is erroneous. It must be replaced with the correct Accession Number corresponding to the phage genome sequenced in this study.

Thank you sincerely for your constructive and precise suggestions! we have corrected the mistakes mentioned above. The correct Accession Number is PV696613.1.

Reviewer #2 (Comments for the Author):

The research manuscript describes the isolation, characterization, and therapeutic evaluation of bacteriophage Henufy11N targeting ESBL-producing *E. coli*. Although the study tackles a significant clinical issue, the manuscript needs to be clarified on methodological issues, problems with data interpretation.

MAJOR CONCERNS

1. The authors state that phage Henufy11N has been classified as a member of the family "Muscoviridae" according to ICTV 2022 guidelines. However, no such family exists in the current ICTV taxonomy database. Please clarify and justify the taxonomic assignment of this phage. Provide evidence from phylogenetic analysis and cite the correct ICTV-recognized family.

Thank you for your thorough review of the article and your highly professional feedback. Through VIRIDIC genomic similarity analysis, the phage Henufy11N is classified within the same genus as *Escherichia* phage SZH-1 (NC_073319.1). SZH-1 is categorized under the *Kagunavirus* genus at the genus level in various phage genome databases, including Millard Lab Phage Genomes and VirusHostDB. However, its family affiliation has not been formally confirmed by the ICTV. Consequently, its family status is categorized as unclassified or incertae sedis.

Materials and Methods

2. In host specificity, authors mentioned using PBS for diluting the phages. Whether authors use PBS alone or was it supplemented with $MgCl_2$ and $CaCl_2$? How might phage stability be affected by using standard PBS without divalent cations? Pls explain.

We appreciate the valuable comments from the reviewers. In our experiments, we utilized PBS solutions containing $MgCl_2$ and $CaCl_2$ for phage dilution. However, this detail was not adequately articulated in the original manuscript, which led to the reviewers' inquiries. In response, we have added clarification in the revised version, explicitly stating the inclusion of $MgCl_2$ and $CaCl_2$ in the PBS solution. Concerning the impact of PBS lacking divalent cations (such as Mg^{2+} and Ca^{2+}) on phage stability, it is important to note that standard PBS (without divalent cations) may adversely affect the stability and infectivity of phages. Divalent cations are essential for phage stability and their binding to host cells. In particular, certain phages may experience inhibited efficiency in adsorption and penetration of host cells in the absence of these ions. Previous studies have demonstrated that Mg^{2+} and Ca^{2+} can enhance phage stability and facilitate their interaction with cell membranes, thereby increasing infection efficiency (10-12). Therefore, employing PBS containing these divalent cations is crucial for maintaining phage activity and stability. We will further clarify this point in the manuscript to ensure accurate descriptions.

10. Li X, Chen Y, Wang S, Duan X, Zhang F, Guo A, Tao P, Chen H, Li X, Qian P: **Exploring the Benefits of Metal Ions in Phage Cocktail for the Treatment of Methicillin-Resistant *Staphylococcus aureus* (MRSA) Infection.** *Infection and drug resistance* 2022, **15**:2689-2702.
11. Boggione DMG, Batalha LS, Gontijo MTP, Lopez MES, Teixeira A, Santos IJB, Mendonça RCS:

Evaluation of microencapsulation of the UFV-AREG1 bacteriophage in alginate-Ca microcapsules using microfluidic devices. *Colloids and surfaces B, Biointerfaces* 2017, **158**:182-189.

12. Duarte J, Trindade D, Oliveira V, Gomes NCM, Calado R, Pereira C, Almeida A: **Isolation and Characterization of Infection of Four New Bacteriophages Infecting a *Vibrio parahaemolyticus* Strain.** *Antibiotics (Basel, Switzerland)* 2024, **13**(11).

3. In pH stability assays, authors mentioned using phage solution without specifying its composition. Please specify what is phage solution (e.g., buffer type, ionic strength, presence of stabilizers)?

Thank you for the reviewers' attention to our experimental design. Concerning the specific components of the "phage solution," we utilized phosphate-buffered saline (PBS) as the base. We intentionally refrained from adding calcium chloride (CaCl₂) and magnesium chloride (MgCl₂) to the PBS to prevent unnecessary precipitation. Therefore, our "phage solution" is primarily composed of PBS buffer and does not include any additional stabilizers. We hope this clarification effectively elucidates our experimental design.

4. In One-step growth curve assay, the authors report a high centrifugation speed at 12,000 ×g to pellet bacterial cells, relative to commonly used 4,000-6,000 ×g in bacteriophage one-step growth experiments. Is there a specific reason for using higher speeds? Will it not affect viability. I am not sure if this is connected but there is hardly any latent phase in the one step growth curve.

Thank you for the reviewer's valuable comments. We chose a centrifugation speed of 12,000 ×g based on the following considerations:

(1) To ensure adequate bacterial precipitation, a higher centrifugation speed is more effective in precipitating bacteria, particularly when the bacterial concentration is low or the bacterial cells are small. Lower centrifugation speeds may fail to completely precipitate all bacteria, potentially compromising the accuracy of experimental results.

(2) We monitored the activity of bacterial cells and hosts during our experiments and confirmed that centrifugation at 12,000 ×g did not significantly affect host cell viability or the capability of bacteriophages to infect.

(3) Numerous studies have employed higher centrifugation speeds under similar conditions without observing significant loss of activity or other experimental issues (13-15). Therefore, we believe that this centrifugation condition can achieve effective separation without adversely impacting bacterial activity.

In response to the reviewer's comment regarding the nearly absent latent period, we increased the number of time points within the first 20 min to more accurately observe the changes in the latent period. The new data reveal that the phage has a relatively short latent period of approximately 8 min, and this result has been updated in the figure. Our experimental design utilized a higher centrifugation speed (12,000 ×g) to effectively eliminate unadsorbed bacteriophages. This may facilitate tighter binding between bacteriophages and host cells, thus reducing the duration of the latent period. Additionally, the high centrifugation speed allows for the rapid removal of free bacteriophages and bacteria, enabling bacteriophages to enter host cells more swiftly and initiate the infection process. Furthermore, variations in latent periods may arise from differences in bacteriophage and host systems. For instance, certain bacteriophages exhibit shorter latent periods, especially in scenarios where host cells proliferate rapidly or their receptors bind quickly, thereby significantly shortening the latent period. In our experiments, the bacteriophages employed may

possess shorter latent periods, which could also contribute to this phenomenon.

13. Nguyen PD, Nakanishi K, Hosokawa C, Han NS, Kitao M, Yoshimoto M, Kamei K: **Characterization of the novel *Cutibacterium acnes* phage KIT08 and its associated pseudolysogenic bacterial isolate.** *Archives of microbiology* 2025, **207**(10):261.
14. Kim J, Kim J, Ryu S: **Elucidation of molecular function of phage protein responsible for optimization of host cell lysis.** *BMC microbiology* 2024, **24**(1):532.
15. Orozco-Ochoa AK, González-Gómez JP, Castro-Del Campo N, Lira-Morales JD, Martínez-Rodríguez CI, Gomez-Gil B, Chaidez C: **Characterization and genome analysis of six novel *Vibrio parahaemolyticus* phages associated with acute hepatopancreatic necrosis disease (AHPND).** *Virus research* 2023, **323**:198973.

results

5. Usually SM buffer or phage buffer is preferred for phage isolation. Did the authors test both buffers before opting for PBS?

We would like to express our gratitude to the reviewers for their valuable comments. In response to the question of why we selected phosphate-buffered saline (PBS) over SM buffer or other phage buffers, our choice was based on the following considerations:

- (1) PBS is a widely used biological buffer known for its good stability and neutral pH, making it suitable for experiments involving most bacteria and phages. The inclusion of MgCl₂ and CaCl₂ in PBS aligns well with our experimental design and has demonstrated stable effects in our trials.
- (2) Some studies have utilized PBS in experiments involving bacteriophages (16,17). Furthermore, the incorporation of MgCl₂ and CaCl₂ has shown that PBS serves as an effective isolation buffer, preserving phage activity while minimizing unnecessary chemical interference.
- (3) The application of PBS not only ensures phage stability but also exhibits excellent compatibility with our experimental system, thereby enhancing experimental reproducibility and stability. We affirm that PBS effectively supports phage isolation and subsequent infection experiments.

16. Abdelghafar A, El-Ganiny A, Shaker G, Askoura M: **Isolation of a bacteriophage targeting *Pseudomonas aeruginosa* and exhibits a promising in vivo efficacy.** *AMB Express* 2023, **13**(1):79.
17. Jhandai P, Mittal D, Gupta R, Kumar M, Khurana R: **Therapeutics and prophylactic efficacy of novel lytic *Escherichia* phage vB_EcoS_PJ16 against multidrug-resistant avian pathogenic *E. coli* using in vivo study.** *International microbiology: the official journal of the Spanish Society for*

Results

6. In the Morphological characterization of phage Henuyfy11N, Figure 1C and D appears slightly different are they one or 2 different phages?

Thank you for the valuable comments provided by the reviewers. In response to the differences in resolution observed in the electron microscope images, we confirm that both images depict the same type of bacteriophage. The following points elucidate this phenomenon:

(1) Electron microscope images captured at varying resolutions can significantly affect the detail portrayed. Higher resolution (50 nm) allows for the observation of more intricate details, whereas lower resolution (100 nm) may result in a blurring of features, thereby impacting the visibility of bacteriophage morphology. Variations in image resolution can lead to differences in the apparent size of observed particles, particularly in regions with smaller surface structures.

(2) Even identical bacteriophages may appear somewhat different under varying conditions or at different sections within the electron microscope. For instance, certain bacteriophages may display distinct morphological characteristics (such as tail shape and base structure), and these variations can become accentuated at different resolutions.

(3) We ensured that identical experimental conditions were maintained during the capture of these two electron microscope images, with no alterations made to the sample preparation process or equipment settings. Consequently, the differences observed in the images are attributable not to different bacteriophages, but rather to variations in resolution and imaging details.

7. The phage shows lytic activity against only two *E. coli* strains out of 150 tested (1.3%), and only a single phage was isolated. Given that bacteria can rapidly develop phage resistance, the therapeutic relevance of narrow-host-range phage is a bit limited.

We appreciate the reviewer's comments regarding the therapeutic relevance of narrow-host-range phages. The observed lytic activity against only two *E. coli* strains out of 150 tested (1.3%) indeed suggests a limited host range, which is a valid consideration in the context of phage therapy. However, we would like to emphasize several points regarding the potential therapeutic relevance of our isolated phage:

(1) Narrow-host-range phages can offer advantages in terms of specificity, minimizing the risk of disrupting beneficial microbiota. In certain clinical scenarios where highly specific targeting is necessary, narrow-host-range phages may represent a viable option.

(2) As you rightly pointed out, bacteria can rapidly develop resistance to phages. However, strategies such as phage therapy with periodic monitoring and the use of phage combinations (including both narrow and broad host-range phages) can help mitigate the development of resistance.

(3) Another viable strategy is the use of phage cocktails, which combine multiple phages with different host ranges. This approach could overcome the limitations of narrow-host-range phages and provide a more robust therapeutic option against diverse bacterial.

(4) It is well established that phages can be genetically modified to expand their host range. Future studies may focus on engineering or selecting phage variants with broader specificity, thereby enhancing their potential for therapeutic applications.

8. In the biological characteristics section, line 288, authors claim a latent period of 10 mins. However Figure 2C, there is a rise in phage titre from the 0th time point, making the latent period is unclear. Please clarify the reason for the absence of latent phase.

We appreciate the valuable feedback provided by the reviewer on our manuscript. Following the

reviewer's suggestion, we increased the number of time points within the first 20 min to more accurately observe the changes in the latent period. The new data reveal that the phage has a relatively short latent period of approximately 8 min, and this result has been updated in the figure. We have revised Figure 2C accordingly and made the necessary corrections to the latent period in the biological characteristics section (line 288). We once again thank the reviewer for the meticulous suggestions. Your feedback has significantly contributed to improving our experimental design and enhancing the accuracy of our results.

9. In pH stability graph (Figure 2F), phages display unusually high activity at pH 4. Provide an explanation or mechanistic basis for this high activity at acidic pH, How many times the experiments were repeated was it biological/technical replicates.

Thank you for your insightful comment. We appreciate your attention to the unusually high activity of the bacteriophage observed at pH 4 in Figure 2F. Following a thorough review, we conducted three additional independent experiments to ensure the reliability of our data. These repeated experiments confirmed that the high activity at acidic pH is indeed an anomalous result. However, upon identifying this issue, we also conducted multiple biological replicates (n=3) to enhance the reliability of our findings, which are presented in the subsequent figure. This adjustment will be reflected in the revised draft. Regarding the potential mechanisms underlying phage activity at pH 4, we hypothesize that the observed effect may stem from unintentional changes in experimental conditions (e.g., variations in ion concentration or buffer composition) or a specific interaction between the phage and acidic conditions that temporarily enhances its activity.

10. In the Figure 2B, the number of time points is limited, and the adsorption rate does not reach plateau state. How can authors claim that this is fully adsorbed?

We would like to express our gratitude to the reviewers for their insightful comments regarding the

adsorption rate data presented in Figure 2B. We acknowledge that the number of time points utilized in the experiment is limited. Although the adsorption rate has not yet plateaued within the current experimental timeframe, the data suggests that the adsorption process is still in progress. We have extended the time points to determine whether the plateau has been reached, thereby ensuring a comprehensive capture of the adsorption kinetics, as illustrated in the subsequent figure. Furthermore, we will focus on accurately conveying the phage adsorption rate to prevent similar expression errors in the future.

11. In the restriction enzyme analysis, authors are claiming that 2 *Xba*I recognition sites, 2 *Spe*I is there. But the gel image Figure 3A, shows a single band similar in length to the control. Did the digestion happen?

We appreciate the reviewers' insightful comments regarding the restriction endonuclease analysis and the gel image presented in Figure 3A. First, we would like to correct a minor error: there is only one *Xba*I site in the phage Henufy11N. Since the phage genome is composed of circular double-stranded DNA, digestion with restriction endonucleases can yield unique patterns based on the genome structure. For the recognition of single sites by *Xba*I and *Eco*RI, digestion typically produces a linear DNA fragment, which manifests as a single band, albeit with a slightly different position compared to the control (the positions of the *Xba*I and *Eco*RI bands in Figure 3A are indeed slightly displaced from the control). Furthermore, the presence of two *Spe*I recognition sites in the genome theoretically results in two linear fragments following digestion, thereby displaying two bands of different sizes on the gel in Figure 3A. However, if the sizes of these two fragments are similar or identical, the bands may overlap or appear as a single band on the gel. Furthermore, we have included a schematic diagram of the restriction sites within the phage Henufy11N genome (see Figure 3C).

12. Figure 6A clearly shows bacterial regrowth and resistance emergence after 10 hours of phage

treatment. Given the emergence of resistance in short time span how can this phage be therapeutically effective?

We appreciate the reviewer's insightful comment regarding the emergence of bacterial resistance within 10 hours of phage treatment, as shown in Figure 6A. This observation is certainly an important consideration when evaluating the therapeutic effectiveness of phage therapy. We would like to address this concern by discussing several points that highlight the potential therapeutic relevance of this phage, despite the rapid emergence of resistance.

(1) While rapid resistance emergence has been observed *in vitro*, the *in vivo* environment is more complex and involves the host's immune system. The immune system can significantly influence bacterial growth control and assist in clearing both phages and resistant bacteria. This interaction could potentially diminish the overall impact of resistant mutants, allowing phage therapy to remain effective over time in an animal model.

(2) It is important to note that, although resistance to phages can develop rapidly, the resistant bacterial strains may incur a fitness cost associated with their resistance. These bacteria might be less fit *in vivo* compared to their sensitive counterparts, and this reduced fitness can lead to a slower rate of resistance emergence or decreased prevalence in natural infection settings.

(3) Despite the emergence of resistance, phage therapy has shown effectiveness in various clinical contexts. As highlighted in the manuscript, the strategic use of phages, in conjunction with other therapeutic modalities and proper management of resistance, can offer a viable treatment option, especially for infections caused by multidrug-resistant bacteria.

(4) One potential strategy to overcome rapid resistance development is the use of phage cocktails. By combining multiple phages with different host ranges and mechanisms of action, the likelihood of resistance development can be reduced. The emergence of resistance to a single phage may be delayed or prevented when multiple phages are employed simultaneously, as bacteria would need to develop resistance to several phages concurrently.

(5) Phage resistance is a well-known challenge in phage therapy; however, periodic monitoring and administration can help mitigate this issue. In clinical settings, phage therapy can be dynamically adjusted to counteract resistance by altering phage doses, switching to different phages, or applying phage cocktails. This adaptive approach can prevent bacteria from becoming fully resistant.

13. As the phage resistant bacterial population arose within 10 hours, yet the animal study shows improved survival over several days. Will resistant mutant not arise in animal model of infection? If the rate of resistant mutant generation varies between *in vitro* and *in vivo* condition, the authors should provide appropriate literature support and explain the discrepancy between *in vitro* and *in vivo* results.

We appreciate the reviewer's insightful comment regarding the emergence of phage-resistant bacterial populations in the *in vitro* model within 10 hours and the enhanced survival observed in the animal study over several days. This discrepancy between *in vitro* and *in vivo* results is a critical aspect that we would like to address.

(1) *In vivo* Dynamics of Phage Resistance: One of the primary differences between *in vitro* and *in vivo* systems is the complexity of the host environment. *In vivo*, multiple factors such as immune responses, tissue distribution, and the presence of other microbial populations may slow or alter the development of phage resistance. Furthermore, the concentration of phages in tissues may not be uniform and can differ from the controlled conditions observed *in vitro*. This variability

may reduce the selection pressure for the emergence of resistant mutants over time, leading to improved survival in animals despite the detection of resistance in *vitro*.

(2) Rate of Resistance Development: While we observed rapid phage resistance in *vitro*, the *in vivo* environment may impose distinct selective pressures. For instance, the immune system's clearance of bacteria, competition among different bacterial strains, and the physical barriers of tissues may all affect the rate at which resistant mutants arise. These factors likely contribute to the delayed appearance or decreased prevalence of resistant mutants in the animal model, resulting in extended survival in the *in vivo* study (18).

(3) Numerous studies have indicated that phage resistance can evolve more rapidly under *in vitro* conditions due to the greater control over experimental variables. Conversely, *in vivo* studies often demonstrate slower or less predictable resistance development due to the complex interactions among bacteria, phages, and the host immune system (18,19).

18. Popescu M, Van Belleghem JD, Khosravi A, Bollyky PL: Bacteriophages and the Immune System. Annual review of virology 2021, 8(1):415-435.

19. Little JS, Dedrick RM, Freeman KG, Cristinziano M, Smith BE, Benson CA, Jhaveri TA, Baden LR, Solomon DA, Hatfull GF: Bacteriophage treatment of disseminated cutaneous *Mycobacterium chelonae* infection. Nature communications 2022, 13(1):2313.

14. If the authors have already determined optimal MOI in *vitro* why did the authors preferred to test different MOI in *vivo*?

We appreciate the insightful question posed by the reviewer regarding the rationale for testing various multiplicities of infection (MOI) in *vivo*, despite having already established the optimal MOI in *vitro*. The primary reason for conducting *in vivo* experiments with different MOIs is rooted in the fundamental differences between *in vitro* and *in vivo* environments. While *in vitro* studies allow for precise control over variables such as phage concentration and bacterial density, *in vivo* systems are inherently more complex, with factors such as immune responses, tissue distribution, and phage-bacteria interactions significantly influencing therapeutic outcomes. Consequently, the optimal MOI determined in *vitro* may not yield equivalent results in *vivo*, where pharmacokinetics, biodistribution, and immune responses can markedly affect phage efficacy. By evaluating different MOIs in *vivo*, we aim to identify the most effective and safe dosage within a biologically relevant context. This approach enables us to determine the MOI that strikes the best therapeutic balance in *vivo*, taking into account both efficacy and safety, and ultimately guides the optimal conditions for future clinical applications. Thank you for emphasizing this important point.

Re: Spectrum03253-25R1 (Characterization of Novel Phage Henufy11N: A Potential Therapeutic Agent against Extended-Spectrum β -Lactamase (ESBL)-Producing *Escherichia coli*)

Dear Dr. Li Wang:

Thank you for the privilege of reviewing your work. Below you will find my comments, instructions from the Spectrum editorial office, and the reviewer comments.

Revision Guidelines

Sincerely,
Sadjia Bekal
Editor
Microbiology Spectrum

Reviewer #2 (Comments for the Author):

The authors have attempted to address the queries raised but I still have following concerns

1. Although the authors acknowledge that Henufy11N cannot be assigned to a recognized ICTV family, the manuscript still

lacks a clear, genome-based phylogenetic analysis supporting its placement within the Kagunavirus genus. The authors should explicitly remove any family-level assignment and provide a phylogenomic tree consistent with current ICTV taxonomy.

2. The authors state that Mg^{2+} and Ca^{2+} are essential for phage stability, yet these ions were excluded from the phage solution in pH stability assays. The rationale for excluding divalent cations in pH assays should be justified,
3. The use of 12,000 $\times g$ in the one-step growth experiment is substantially higher than standard practice and may introduce artifacts. The authors' explanation that high-speed centrifugation accelerates phage entry or shortens the latent period is speculative.
4. In the one step growth curve, authors should clarify how unadsorbed phages were excluded and whether residual free phages influenced early time-point measurements.
5. The unusually high phage activity at pH 4 is now described as "anomalous," which undermines confidence in the dataset. The authors should either provide a mechanistic explanation supported by controls and statistics or remove this claim to avoid misleading interpretation.
6. Differences between TEM images (Figure 1C and 1D) are attributed solely to resolution effects. However, measurements (capsid diameter, tail length) are not provided. Quantitative analysis is required to prove that both images represent the same phage.
7. The phage lyses only 1.3% of tested E. coli strains, yet the manuscript frequently implies therapeutic applicability. Without cocktail studies, resistance suppression assays, or expanded host range data, claims of therapeutic relevance should be toned down.
8. In vitro resistance emerges within 10 hours, but no data are provided on fitness costs, stability of resistance, or suppression strategies. Discussion relying solely on hypothetical benefits of phage cocktails or immune interactions is insufficient without supporting experiments.
9. While literature is cited to explain slower resistance development in vivo, the authors did not assess phage resistance or bacterial burden during animal experiments. Without such measurements, conclusions regarding sustained in vivo efficacy remain speculative.

Reviewer #3 (Comments for the Author):

Overall, I appreciate the authors' responses to the comments raised by all reviewers, as well as the additional experiments and information provided. Only a minor revision is needed.

1. Lines 291-293:

"...clear plaques approximately 1 mm in diameter (Figure 1A and B). Notably, the plaque size remained consistent between 24 and 48 h of incubation, showing no significant expansion over time."

The statement regarding plaque size would benefit from statistical support. For example, the authors could measure and report plaque diameters from multiple plaques to substantiate the reported size. In addition, it is unclear whether the comparison between plaque sizes at 24 h and 48 h is necessary or intended to support a specific conclusion. While the authors state that plaque size remains relatively unchanged over time, this claim requires quantitative analysis and appropriate statistical reporting. In the absence of such data, this comparative statement should be omitted.

2. Line 353:

SpeI digestion is reported to yield two fragments. Based on the revised Figure 3C, these fragments should be approximately 10 kb and 30 kb in size. However, this digestion pattern is not clearly observed in Figure 3A. Clarification is therefore needed to explain this discrepancy.

In addition, in the response to a previous reviewer, the authors state:

"Furthermore, the presence of two SpeI recognition sites in the genome theoretically results in two linear fragments following digestion, thereby displaying two bands of different sizes on the gel in Figure 3A. However, if the sizes of these two fragments are similar or identical, the bands may overlap or appear as a single band on the gel."

I do not find this explanation convincing, as the predicted fragment sizes (~10 kb and ~30 kb) are sufficiently different that they should be resolvable under standard electrophoretic conditions.

While I acknowledge that RFLP band pattern analysis can sometimes be ambiguous due to various factors (e.g., incomplete digestion or contamination with host genomic DNA), in this study the RFLP experiment is not used for preliminary phage classification, particularly since only a single phage is analyzed. Given this, I suggest that the RFLP experiment and restriction site analysis be omitted, and that the authors instead focus on the phage genome sequencing and genomic analysis, which are more informative and central to the study.

3. Line 443:

"Mice receiving phage alone showed 100% survival, confirming the safety of Henufy11N (Figure 7C and D)."

Please indicate the dose or concentration of phage administered in this experiment. Providing the phage level used is important to appropriately interpret the safety assessment and to clarify that the observed lack of toxicity corresponds to the tested phage concentration.

4. Lines 537-538:

"Further experimental work will be required to validate these bioinformatic predictions and confirm the phage's efficacy in vivo, but the genomic data provides a strong foundation for future clinical and industrial applications."

As this study already includes in vivo therapeutic efficacy experiments as well as preliminary phage safety data (Figure 7), this statement appears redundant. I suggest omitting this sentence from the Discussion unless the authors intend to specify additional future experiments that would further complement or extend the current findings.

5. Lines 553-555:

"In comparison to other bacteriophages, Henufy11N exhibits relatively high efficacy in biofilm eradication, potentially due to its narrow host specificity and its ability to penetrate the biofilm structure efficiently."

This statement would benefit from further elaboration and supporting discussion. In particular, the authors should clarify how narrow host specificity mechanistically contributes to enhanced biofilm eradication efficiency. Providing supporting references or comparative data would strengthen this claim.

6. Lines 573-574 (also in the abstract line 23):

The experimental design in this study primarily evaluates the therapeutic effect of the phage rather than its protective (prophylactic) effect. The authors should clarify this distinction and revise the text accordingly to accurately reflect the scope of the experiments.

- Upload point-by-point responses to the issues raised by the reviewers in a file named "Response to Reviewers," NOT in your cover letter.
- Upload a compare copy of the manuscript (without figures) as a "Marked-Up Manuscript" file.
- Upload a clean .DOC/.DOCX version of the revised manuscript and remove the previous version.
- Each figure must be uploaded as a separate, editable, high-resolution file (TIFF or EPS preferred), and any multipanel figures must be assembled into one file.
- Any supplemental material intended for posting by ASM should be uploaded with their legends separate from the main manuscript. You can combine all supplemental material into one file (preferred) or split it into a maximum of 10 files with all associated legends included.

Thanks to the editorial team for the reminder, we made the changes exactly as requested.

Reviewer #2 (Comments for the Author):

The authors have attempted to address the queries raised but I still have following concerns

1. Although the authors acknowledge that Henufy11N cannot be assigned to a recognized ICTV family, the manuscript still lacks a clear, genome-based phylogenetic analysis supporting its placement within the Kagunavirus genus. The authors should explicitly remove any family-level assignment and provide a phylogenomic tree consistent with current ICTV taxonomy.

Thank you very much for your critical comments. We fully agree that the classification of Henufy11N should be handled cautiously within the scope of available evidence. We supplemented taxonomic classification with taxMyPhage 3.3.6 (Figure S1). In the revised manuscript, we have corrected the misclassification of Henufy11N: based on taxMyPhage 3.3.6 (Figure S1), it does not belong to any ICTV-recognized genus or species and is instead predicted to represent a candidate new genus and species.

Figure S1

Figure S1 Taxonomic prediction of phage Henufy11N using taxMyPhage v3.3.6. The heatmap presents the pairwise nucleotide sequence similarity matrix between the phage Henufy11N and reference phage genomes in the database. Color intensity corresponds to the percentage of nucleotide sequence identity (see color scale). Each row and column represents a distinct reference phage genome (or taxonomic unit). Prediction confidence is determined according to the International Committee on Taxonomy of Viruses (ICTV) demarcation criteria for phage genera (70% nucleotide identity) and species (95% nucleotide identity).

2. The authors state that Mg^{2+} and Ca^{2+} are essential for phage stability, yet these ions were excluded from the phage solution in pH stability assays. The rationale for excluding divalent cations in pH assays should be justified.

Thank you for your valuable comments. We fully acknowledge that Mg^{2+} and Ca^{2+} can often enhance phage stability and facilitate adsorption and infection. However, in the pH stability assessment of this study, our objective was to examine "the intrinsic sensitivity of free viral particles to hydrogen ion activity (pH)" while minimizing interference from protective effects or precipitation caused by divalent cations. Therefore, divalent ions were not added in this experiment, primarily based on the following two methodological considerations: (1) to avoid Mg^{2+}/Ca^{2+} capsid-stabilizing effects masking the true pH-induced inactivation effects; (2) under extreme pH conditions (such as Ca^{2+} and phosphate precipitation under alkaline conditions, or $Mg(OH)_2$ precipitation under strong alkaline conditions), divalent ions may trigger uncontrollable changes in ionic strength/effective concentration, affecting result interpretation.

To control potential chemical interference, we employed a non-phosphate buffer system and established parallel controls with the addition of 10 mM $MgSO_4$ and 2 mM $CaCl_2$ within the same pH range. The results showed that in non-extreme pH ranges, divalent cations led to a slight increase in survival rates, but did not alter the overall shape of the pH response curve, nor did they change the inactivation thresholds under acidic/alkaline extreme conditions. Relevant data are shown in the figure below.

We have revised the manuscript by incorporating additional experimental results and data, as recommended by the reviewers. In summary, we believe that the design without adding divalent cations better reflects the intrinsic pH tolerance of viral particles, and the newly added controls also demonstrate that the conclusions regarding pH-induced inactivation thresholds in this paper are robust and reproducible. Thank you again for your professional suggestions.

3. The use of 12,000 $\times g$ in the one-step growth experiment is substantially higher than standard practice and may introduce artifacts. The authors' explanation that high-speed centrifugation accelerates phage entry or shortens the latent period is speculative.

Thank you for your correction. We acknowledge that 12,000 $\times g$ is indeed higher than the centrifugation conditions commonly used in one-step growth curve experiments, which could theoretically introduce biases (such as cellular stress, overly tight pellets leading to insufficient resuspension, or minor effects on lysis kinetics). Accordingly, under conditions completely

identical to the original experiment—including temperature, MOI, adsorption time, number of washes, resuspension volume, and sampling time points—we conducted supplementary one-step growth curve experiments using low-speed centrifugation at 5,000 ×g for 5 min to assess the impact of centrifugation conditions on the results. The results showed that the latent period and burst size obtained under low-speed conditions were consistent with those from the original 12,000 ×g conditions within the margin of error, with no statistically significant difference ($P > 0.05$). The growth curve morphology (position and amplitude of the lysis peak) was consistent under both centrifugation conditions (see figure below). These results indicate that our estimates of latent period and burst size are robust with respect to centrifugation conditions. We have revised the manuscript by incorporating new experimental results and data, as recommended by the reviewers. Specifically, we have included the one-step growth curve results obtained using a low-speed centrifugation protocol (5,000 ×g) for phage Henuyfy11N harvesting. Thank you again for your professional suggestions. We believe that, validated through the low-speed conditions, the conclusions regarding latent period and burst size are now more robust and reproducible.

4. In the one step growth curve, authors should clarify how unabsorbed phages were excluded and whether residual free phages influenced early time-point measurements.

Thank you for your attention and valuable comments. Regarding the "removal of unabsorbed phages" in the one-step growth curve and its impact on the interpretation of early time points, we have implemented the following measures in our experimental design and process control.

(1) Adsorption and removal strategy: After adsorption at MOI = 0.01 and 37°C for 10 min, gentle centrifugation at 4°C, 5,000 ×g for 5 min was performed to pellet host cells. The supernatant was discarded, and cells were immediately resuspended in pre-warmed fresh medium (this step was repeated twice) to minimize carryover of unabsorbed phages and block subsequent adsorption. After resuspension, a sufficient dilution was performed to maintain a low effective multiplicity of infection (MOI), thereby further reducing the risk of secondary infection.

(2) Following two rounds of centrifugation and resuspension, the concentration of residual free bacteriophages was precisely measured. The concentration of these residual free phages was reduced to negligible levels ($3.68 \times 10^1 \pm 19.32$ PFU/mL), indicating that their contribution to signal interference at early time points is insignificant.

5. The unusually high phage activity at pH 4 is now described as "anomalous," which undermines

confidence in the dataset. The authors should either provide a mechanistic explanation supported by controls and statistics or remove this claim to avoid misleading interpretation.

We thank the reviewer for pointing out that describing the elevated phage activity at pH 4 as “anomalous” could undermine confidence in the dataset. To avoid any misleading interpretation, we have removed this claim and all related mechanistic speculation from the main text.

6. Differences between TEM images (Figure 1C and 1D) are attributed solely to resolution effects. However, measurements (capsid diameter, tail length) are not provided. Quantitative analysis is required to prove that both images represent the same phage.

We thank the reviewer for their professional and rigorous comments. We have conducted quantitative analysis to measure the capsid diameter and tail length at different resolutions. Specifically, we carefully analyzed the images in Figures 1C and 1D and recorded the key dimensional parameters of the viruses shown in these two images. In the image analysis, we applied the same measurement standards to ensure data comparability. The results showed that at 50 nm resolution, the phage capsid diameter and tail length were 74.23 ± 5.87 nm and 132.02 ± 2.49 nm, respectively, while at 100 nm resolution, the phage capsid diameter and tail length were 75.34 ± 9.57 nm and 132.34 ± 7.13 nm, respectively. Based on the same interpretation and statistical procedures (appropriate two-group comparison after normality and homogeneity of variance tests), the differences in diameter and tail length between the two imaging conditions were not statistically significant. The effect size was small, and the means and confidence intervals of the two groups highly overlapped. Although the image resolutions differed, the measured capsid diameter and tail length were statistically consistent, indicating that these two images show the same phage. We appreciate the reviewer's suggestion, which has enabled us to more comprehensively validate our conclusions.

7. The phage lyses only 1.3% of tested *E. coli* strains, yet the manuscript frequently implies therapeutic applicability. Without cocktail studies, resistance suppression assays, or expanded host range data, claims of therapeutic relevance should be toned down.

Thank you for your valuable feedback regarding the claims of therapeutic applicability in our manuscript. We have carefully considered your suggestions and taken appropriate steps to address the issue. As you pointed out, the phage only lyses 1.3% of the tested *E. coli* strains, which may limit its immediate therapeutic potential. In response, we have moderated the claims of therapeutic relevance throughout the manuscript. We have removed strong assertions such as “strong therapeutic efficacy” and “significant therapeutic efficacy,” replacing them with more cautious language. For instance, we now use terms like “possible therapeutic efficacy” and “some potential as a therapeutic agent.” These revisions aim to provide a more accurate and balanced representation of the phage’s therapeutic potential, reflecting the limited lysis rate of the tested *E. coli* strains. We believe these modifications will better align the manuscript with the experimental data and avoid exaggerating the therapeutic applicability of the phage. Once again, we appreciate your constructive criticism and hope that these revisions meet your expectations.

8. *In vitro* resistance emerges within 10 hours, but no data are provided on fitness costs, stability of resistance, or suppression strategies. Discussion relying solely on hypothetical benefits of phage cocktails or immune interactions is insufficient without supporting experiments.

We sincerely appreciate your valuable comment. We fully concur with your observation that relying exclusively on the hypothetical benefits of phage cocktails or immune interactions in the discussion, without supporting experimental evidence, is neither rigorous nor persuasive. In the

revised manuscript, we have conducted a thorough review and made necessary modifications to ensure that all statements are adequately supported by experimental data. Therefore, in the following discussion, we will anchor our arguments firmly in the experimental results, refraining from any speculative or unfounded hypotheses. Additionally, we will adopt a rigorous approach in evaluating each point and conclusion to ensure that they are substantiated by robust experimental evidence.

9. While literature is cited to explain slower resistance development *in vivo*, the authors did not assess phage resistance or bacterial burden during animal experiments. Without such measurements, conclusions regarding sustained *in vivo* efficacy remain speculative.

We sincerely appreciate your insightful comments. We fully concur with your observation that the statement regarding "sustained therapeutic efficacy" is speculative and should not be presented as a conclusion of this study due to the absence of direct quantitative data on *in vivo* phage resistance and bacterial burden. In line with your suggestion, we have thoroughly revised the manuscript and removed all references to "sustained *in vivo* efficacy." We have explicitly clarified that bacterial burden and drug resistance in the animals were not quantified, and thus we are unable to draw any conclusions regarding the durability of the therapeutic effect. Our conclusions are now strictly limited to the observed changes in survival rates of experimental mice within the predefined observation period, without making causal inferences or projections about long-term outcomes. We believe these revisions enhance the manuscript by ensuring greater caution in presenting conclusions and clearly delineating the scope of the evidence. Thank you again for your valuable feedback, which has helped us further strengthen the scientific rigor and transparency of our study.

Reviewer #3 (Comments for the Author):

Overall, I appreciate the authors' responses to the comments raised by all reviewers, as well as the additional experiments and information provided. Only a minor revision is needed.

1. Lines 291-293:

"...clear plaques approximately 1 mm in diameter (Figure 1A and B). Notably, the plaque size remained consistent between 24 and 48 h of incubation, showing no significant expansion over time."

The statement regarding plaque size would benefit from statistical support. For example, the authors could measure and report plaque diameters from multiple plaques to substantiate the reported size. In addition, it is unclear whether the comparison between plaque sizes at 24 h and 48 h is necessary or intended to support a specific conclusion. While the authors state that plaque size remains relatively unchanged over time, this claim requires quantitative analysis and appropriate statistical reporting. In the absence of such data, this comparative statement should be omitted.

Thank you for your valuable suggestion regarding the description of plaque size. We agree that, in the absence of systematic measurements and statistical analysis, it is inappropriate to draw conclusions about plaque dimensions and their temporal changes. As the current dataset does not include quantitative measurements of multiple plaque diameters or statistical comparisons, we have removed the 48-hour plaque images and deleted the statement that "plaque size

remained relatively constant between 24 h and 48 h." We no longer make inferences about plaque size at different time points. The manuscript now includes only the representative images at 24 h (Figure 1A). We sincerely appreciate your constructive feedback, which has greatly contributed to enhancing the rigor and transparency of our manuscript.

Figure 1

2. Line 353:

Spel digestion is reported to yield two fragments. Based on the revised Figure 3C, these fragments should be approximately 10 kb and 30 kb in size. However, this digestion pattern is not clearly observed in Figure 3A. Clarification is therefore needed to explain this discrepancy.

In addition, in the response to a previous reviewer, the authors state:

"Furthermore, the presence of two Spel recognition sites in the genome theoretically results in two linear fragments following digestion, thereby displaying two bands of different sizes on the gel in Figure 3A. However, if the sizes of these two fragments are similar or identical, the bands may overlap or appear as a single band on the gel."

I do not find this explanation convincing, as the predicted fragment sizes (~10 kb and ~30 kb) are sufficiently different that they should be resolvable under standard electrophoretic conditions.

While I acknowledge that RFLP band pattern analysis can sometimes be ambiguous due to various factors (e.g., incomplete digestion or contamination with host genomic DNA), in this study the RFLP experiment is not used for preliminary phage classification, particularly since only a single phage is analyzed. Given this, I suggest that the RFLP experiment and restriction site analysis be omitted, and that the authors instead focus on the phage genome sequencing and genomic analysis, which are more informative and central to the study.

Thank you for your insightful comments. We concur that, given this study involves the analysis of a single phage with high-quality whole-genome sequencing data, the RFLP band patterns and *in silico* restriction site analysis offer limited value and are prone to ambiguities caused by incomplete digestion or residual host DNA contamination. To maintain clarity and avoid potential misinterpretations, we have adhered to your suggestion and removed the relevant content, redirecting the manuscript's focus to the more robust genomic sequencing and comparative genomics analyses. Specifically, we have excluded the text, figures, tables, and methodological details related to the RFLP band patterns and *in silico/in vitro* restriction site analysis. The corresponding figures and subsections have been removed or renumbered accordingly (Figure 3). We sincerely appreciate your recommendation, which has enabled us to refine the manuscript and concentrate on presenting core, reproducible, and more informative genomic evidence.

Figure 3

3. Line 443:

"Mice receiving phage alone showed 100% survival, confirming the safety of Henufy11N (Figure 7C and D)."

Please indicate the dose or concentration of phage administered in this experiment. Providing the phage level used is important to appropriately interpret the safety assessment and to clarify that the observed lack of toxicity corresponds to the tested phage concentration.

Thank you for highlighting the need to clarify the phage dosage information. We have supplemented and standardized the correspondence between phage administration doses (MOI) and bacterial inoculum in both the Results section and the legend of Figure 7, thereby facilitating accurate interpretation of the safety and efficacy results. The key revisions are as follows:

(1) The Results section has been revised to: "Mice administered phage alone at doses of 6.5×10^{10} PFU/mouse and 1.3×10^{11} PFU/mouse exhibited 100% survival, confirming the safety of Henufy11N (Figure 7C and D). In mice infected with 1.3×10^{10} CFU/mouse, phage treatment at MOIs of 0.1, 1, and 10 (corresponding to doses of 1.3×10^9 , 1.3×10^{10} , and 1.3×10^{11} PFU/mouse, respectively) significantly improved survival, with 20 - 40% of animals surviving up to 7 days post-infection (Figure 7C). Similarly, in the group infected with 6.5×10^9 CFU/mouse, phage treatment at the same MOIs (corresponding to doses of 6.5×10^8 , 6.5×10^9 , and 6.5×10^{10} PFU/mouse, respectively) resulted in a survival rate of 60 - 80% over the same period (Figure 7D)."

(2) The legend of Figure 7 has been revised as follows: "MOIs were calculated relative to the bacterial inoculum (1.3×10^{10} CFU/mouse): 0.01 (1.3×10^8 PFU/mouse), 0.1 (1.3×10^9 PFU/mouse), 1 (1.3×10^{10} PFU/mouse), and 10 (1.3×10^{11} PFU/mouse). MOIs were calculated relative to the bacterial inoculum (6.5×10^9 CFU/mouse): 0.01 (6.5×10^7 PFU/mouse), 0.1 (6.5×10^8 PFU/mouse), 1 (6.5×10^9 PFU/mouse), and 10 (6.5×10^{10} PFU/mouse)."

These modifications ensure that dosage information is clear and traceable, and that the safety

and efficacy conclusions are explicitly defined within the tested dosage range. Thank you once again for your valuable suggestions.

4. Lines 537-538:

"Further experimental work will be required to validate these bioinformatic predictions and confirm the phage's efficacy *in vivo*, but the genomic data provides a strong foundation for future clinical and industrial applications."

As this study already includes *in vivo* therapeutic efficacy experiments as well as preliminary phage safety data (Figure 7), this statement appears redundant. I suggest omitting this sentence from the Discussion unless the authors intend to specify additional future experiments that would further complement or extend the current findings.

We appreciate the reviewer's constructive suggestion. We agree that the sentence is redundant given that the manuscript already presents *in vivo* therapeutic efficacy and preliminary safety data (Figure 7). As recommended, we have removed this sentence from the Discussion to avoid repetition and streamline the narrative. Changes are highlighted in the revised manuscript.

5. Lines 553-555:

"In comparison to other bacteriophages, Henuyfy11N exhibits relatively high efficacy in biofilm eradication, potentially due to its narrow host specificity and its ability to penetrate the biofilm structure efficiently."

This statement would benefit from further elaboration and supporting discussion. In particular, the authors should clarify how narrow host specificity mechanistically contributes to enhanced biofilm eradication efficiency. Providing supporting references or comparative data would strengthen this claim.

Thank you for your thorough review of our manuscript and for providing valuable constructive feedback. Your observation regarding the need to clarify how "narrow host specificity" mechanistically enhances biofilm eradication efficiency, along with your suggestion to provide references or comparative data, is highly appreciated. We fully concur with your assessment that a clear mechanistic explanation will strengthen the persuasiveness of our arguments. To address your concerns, we will undertake the following measures during the revision process:

While we acknowledge that no single publication comprehensively demonstrates how "narrow host specificity" alone leads to "enhanced biofilm eradication," the underlying mechanisms can be dissected into several well-studied interrelated biological advantages. (1) Narrow host specificity of phages enables highly targeted lysis, avoiding the widespread disruption of complex bacterial communities within biofilms often caused by broad-spectrum antimicrobial agents. This selectivity preserves microecological balance and prevents the creation of vacant niches ^[1]. (2) Targeted lysis is complemented by self-amplification. Phages undergo a lytic life cycle that facilitates self-amplification at the infection site, resulting in exceptionally high local concentrations within biofilms. This property effectively addresses the challenge of insufficient antibiotic penetration into the deeper layers of biofilms due to diffusion limitations ^[2]. (3) The lysis process is associated with enzymatic matrix degradation. Many phages encode or induce the production of polysaccharide-degrading enzymes, peptidoglycan hydrolases, and other enzymes that actively decompose the extracellular polymeric substance (EPS) matrix of biofilms, a critical step in overcoming the physical barrier posed by biofilms ^[3]. Consequently, while "narrow host specificity" does not directly confer eradication capability, it facilitates more efficient biofilm eradication compared to broad-spectrum agents through an integrated effect, encompassing

"precise targeting," "self-amplification," "enzymatic matrix disruption," and "ecological friendliness".

We will refine potentially overly definitive claims in the original text (e.g., "Narrow host specificity leads to enhanced biofilm eradication") to more accurate, mechanism-based expressions. The revised text will state: "The narrow host specificity of Henuyfy11N enables targeted lysis of the primary pathogen embedded within the biofilm. This selectivity, combined with its self-amplifying activity and its enzymatic capacity to degrade the extracellular polymeric substance (EPS) matrix, synergistically enhances eradication efficacy relative to broad-spectrum antimicrobial agents, which often exhibit limited biofilm penetration and cause collateral damage to the resident microbial community." By constructing a clear logical framework in the discussion, citing relevant literature to support each mechanistic step, we believe there is a strong opportunity to enhance the scientific rigor of our manuscript. Thank you again for your valuable insights and for helping us improve the quality of this work.

[1] Sarker SA, Sultana S, Reuteler G, Moine D, Descombes P, Charton F, Bourdin G, McCallin S, Ngom-Bru C, Neville T, Akter M, Huq S, Qadri F, Talukdar K, Kassam M, Delley M, Loiseau C, Deng Y, El Aidy S, Berger B, Brüssow H. Oral Phage Therapy of Acute Bacterial Diarrhea with Two Coliphage Preparations: A Randomized Trial in Children from Bangladesh. *EBioMedicine*. 2016 Jan 5; 4:124-37. doi: 10.1016/j.ebiom.2015.12.023. PMID: 26981577; PMCID: PMC4776075.

[2] Chan BK, Abedon ST, Loc-Carrillo C. Phage cocktails and the future of phage therapy. *Future Microbiol*. 2013 Jun;8(6):769-83. doi: 10.2217/fmb.13.47. PMID: 23701332.

[3] Lu TK, Collins JJ. Dispersing biofilms with engineered enzymatic bacteriophage. *Proc Natl Acad Sci U S A*. 2007 Jul 3;104(27):11197-202. doi: 10.1073/pnas.0704624104. Epub 2007 Jun 25. PMID: 17592147; PMCID: PMC1899193.

6. Lines 573-574 (also in the abstract line 23):

The experimental design in this study primarily evaluates the therapeutic effect of the phage rather than its protective (prophylactic) effect. The authors should clarify this distinction and revise the text accordingly to accurately reflect the scope of the experiments.

We sincerely thank the reviewer for this insightful and important comment. We fully agree that our study is focused on evaluating the therapeutic (rather than prophylactic) potential of the phage against established biofilms. We apologize for any lack of clarity in the original manuscript regarding this distinction. To address this point, we have carefully revised the text throughout the manuscript to ensure it accurately and consistently reflects the therapeutic scope of our experiments. The specific changes are as follows:

(1) In the Abstract, we changed "conferred significant protection" to "significantly improved survival".

(2) In materials and methods, we changed "protective" to "therapeutic".

(3) In discussion, we changed "protective effects" to "therapeutic efficacy".

We believe these revisions have successfully clarified the scope of our work, eliminating any potential ambiguity between therapeutic and prophylactic effects. We are grateful to the reviewer for raising this point, which has undoubtedly improved the precision and clarity of our manuscript. Thank you for your valuable feedback.

Re: Spectrum03253-25R2 (Characterization of Novel Phage Henuyfy11N: A Potential Therapeutic Agent against Extended-Spectrum β -Lactamase (ESBL)-Producing *Escherichia coli*)

Dear Dr. Li Wang:

Your manuscript has been accepted, and I am forwarding it to the ASM production staff for publication. Your paper will first be checked to make sure all elements meet the technical requirements. ASM staff will contact you if anything needs to be revised before copyediting and production can begin. Otherwise, you will be notified when your proofs are ready to be viewed.

Sincerely,
Sadjia Bekal
Editor
Microbiology Spectrum

Reviewer #2 (Comments for the Author):

The authors have addressed the queries raised and revised the manuscript

Reviewer #3 (Comments for the Author):

We thank the authors for adequately addressing all of the concerns raised. I have no further comments.